# Neural signaling contributes to heart formation and growth in the invertebrate chordate, *Ciona robusta*

Hannah N. Gruner[1☉], C. J. Pickett[1☉], Jasmine Yimeng Bao[1], Richard Garcia[1], Akiko Hozumi[2], Tal D. Scully[3], Sydney Popsuj[1], Shaoyang Ning[4], Mavis Gao[1], Gia Bautista[1], Keren Maze[1], HaEun Karissa Lim[1], Tomohiro Osugi[5], Mae Collins-Doijode[1], Ofubofu Cairns[1], Gabriel Levis[1], Shu Yi Chen[1], TaiXi Gong[1], Honoo Satake[5], Allon Moshe Klein[3], Yasunori Sasakura[2], Bradley Davidson[1]*

**1** Department of Biology, Swarthmore College, Swarthmore, Pennsylvania, United States of America, **2** Shimoda Marine Research Center, University of Tsukuba, Shizuoka, Japan, **3** Department of Systems Biology, Harvard Medical School, Boston, Massachusetts, United States of America, **4** Department of Mathematics and Statistics, Swarthmore College, Swarthmore, Pennsylvania, United States of America, **5** The Bioorganic Research Institute, Suntory Foundation for Life Sciences (SUNBOR), Kyoto, Japan

☉ These authors contributed equally to this work.
* bdavids1@swarthmore.edu

## Abstract

Neurons contribute to the complex interplay of signals that mediate heart development and homeostasis. Although a limited set of studies suggest that neuronal peptides impact vertebrate heart growth, the specific contributions of these peptides to cardiomyocyte progenitor differentiation or proliferation have not been elucidated. Here, we show that the neuropeptide tachykinin along with canonical Wnt signaling regulate cardiomyocyte progenitor proliferation in the chordate model *Ciona robusta*. In *C. robusta*, the heart continues to grow throughout adulthood and classic histological studies indicate that a line of undifferentiated cells may serve as a reserve progenitor lineage. We found that this line of cardiomyocyte progenitors consists of distinct distal and midline populations. Our analysis indicates that distal progenitors divide asymmetrically to produce distal and midline daughters while midline progenitors divide asymmetrically to produce myocardial precursors. Through single-cell RNA sequencing (scRNA-seq) of adult *C. robusta* hearts, we delineated the cardiomyocyte progenitor expression profile. Based on this data, we investigated the role of Wnt signaling in cardiomyocyte progenitor proliferation and found that canonical Wnt signaling is required to suppress excessive progenitor proliferation. The scRNA-seq data also identified a number of presumptive cardiac neural-like cells. Strikingly, we found that a subset of these neuronal cells appears to innervate the distal cardiomyocyte progenitors. Based on tachykinin receptor expression in these neural-like cells, we blocked tachykinin signaling using pharmacological inhibitors and found that this led to reduced proliferation in the distal progenitor pool. Through targeted CRISPR-Cas9 knockdown, we then demonstrated that both extrinsic tachykinin and

**Data availability statement:** Adult Ciona heart 10× single cell sequencing FASTQ files are available on the NCBI Sequence Read Archive (SRA). Both replicates are found in BioProject accession PRJNA1420413. SRR37160362 corresponds to heart sample 1, and SRR37288072 corresponds to heart sample 2. Additional data corresponding to various graphs can be found in supplemental file S1 Data.

**Funding:** This work was supported by two grants received by B.D. (The American Heart Association grant number 20AIREA35080013 (https://professional.heart.org/en/research-programs/aha-funding-opportunities) and The National Science Foundation grant number 8077804 (https://www.nsf.gov/)), and the National Institutes of Health Ruth L. Kirschstein Postdoctoral Individual National Research Service Award (1F32HL170997) fellowship supporting H.N.G. The funders had no role in study design, data collection and analysis, decision to publish, or preparation of the manuscript.

**Competing interests:** The authors have declared that no competing interests exist.

**Abbreviations:** BIO, 6-Bromoindirubin-3′-oxime; BMP, morphogenetic protein; cWnt, canonical Wnt; DBH, dopamine beta-hydroxylase; ECM, extracellular matrix; FGF, fibroblast growth factor; FSW, filtered artificial seawater; IGF, insulin growth factor; PCA, principal component analysis; PC2, Prohormone Convertase 2; RA, retinoic acid; scRNA-seq, single-cell RNA sequencing; SHP, second heart precursor; TH, tyrosine hydroxylase; UL, undifferentiated line; UMAP, Uniform Manifold Approximation and Projection; Wnt, wingless-related integration site.

intrinsic cardiac tachykinin receptors are required for formation of the myocardial heart tube. This work provides valuable insights regarding the deployment of neural signals to regulate organ growth in response to environmental or homeostatic inputs.

## Introduction

### Neural signaling and development

Neurons contribute to the complex interplay of signals that regulate cellular behaviors during development. Studies in a wide range of organisms have shown that both neurotransmitters and neuropeptides impact fate specification or behavior of neural lineage cells along with a wide range of non-neural lineages [1–22]. A limited set of studies suggest that neuronal signals regulate vertebrate heart growth [23].

### Vertebrate heart growth

Vertebrate heart growth has been most intensively studied in mammals, particularly in mice [24–26]. Human and mouse embryonic hearts exhibit extremely high levels of growth during cardiogenic patterning stages [27,28], peaking during the period of chamber expansion when heart volume increases by over 100-fold [29–31]. The period of extensive growth also continues after morphogenesis is complete. Throughout murine embryonic and fetal stages there are similar increases in myocardial volume and cardiomyocyte numbers indicating that heart growth is largely driven by proliferation [30]. Recent studies, including extensive single-cell RNA sequencing (scRNA-seq) assays, have revealed unanticipated levels of heterogeneity within early progenitor populations and mapped some of these distinct progenitors within the heart field [32–38]. Studies that encompass later stages of cardiogenic patterning have shown that early progenitor pools persist at the venous and arterial poles where they contribute to heart tube elongation [27,33,39–45].

### Signals that regulate heart growth

Extensive research in both vertebrates and fruit flies has documented signal-dependent coordination of initial cardiogenic patterning along with homeostasis and regeneration of the mature organ [46–55]. In contrast, relatively few studies have examined signal-dependent regulation of heart growth during the intermediate stage following completion of morphogenesis [24,25,27]. Numerous signaling pathways coordinate heart growth by modulating either cardiomyocyte specification or proliferation. These include the fibroblast growth factor (FGF), insulin growth factor (IGF), hedgehog, wingless-related integration site (Wnt), bone morphogenetic protein (BMP), retinoic acid (RA), hippo and notch pathways along with microenvironmental cues associated with the extracellular matrix (ECM) such as hyaluronic acid [53,55–67]. Some of these pathways, such as Wnt and FGF, have extensive, cascading roles in myocardial patterning and growth that have been difficult to disentangle [53,63,64,67–69]. Recent scRNA sequencing studies have delineated precise spatiotemporal expression patterns of receptors and ligands that are potentially involved in

myocardial growth [34]. Predicted receptor/ligand interactions from these studies align with previous functional data for a number of signals including IGF, BMP, and Notch while also revealing new candidate pathways [32–34]. However, these studies have not focused on the potential contribution of neural signals to heart growth.

## The impact of neuronal signals on vertebrate heart development and growth

The vertebrate heart is innervated by extrinsic neurons from the central nervous system along with intrinsic cardiac neurons. Cardiac intrinsic plexuses regulate heart rate and rhythm largely through modulation of the cardiac conduction system. The cardiac conduction system is composed of specialized cardiomyocytes that function as slow-conducting pacemaker cells in the sinoatrial and atrioventricular nodes along with fast-conducting cells in structures such as the Bundle of His and Purkinje fiber network [70,71]. The vagus nerve is the primary source of extrinsic cardiac innervation and plays a well-defined role in modulating intrinsic plexuses and cardiac conduction [72,73]. In mouse embryos, extensions of the vagal nerve first innervate the embryonic heart during a period of dramatic proliferative growth following completion of morphogenesis [74–76]. A number of studies indicate that innervation promotes cardiomyocyte maturation [77,78]. However, studies regarding the potential role for neuronal signals in vertebrate cardiomyocyte proliferation have largely focused on regeneration or homeostasis [2,23,79,80]. A more limited set of studies have investigated the impact of both neurotransmitters and neuropeptides on cardiomyocyte proliferation during earlier developmental stages [77,81,82]. One of these studies indicates that vagus nerve neuropeptide secretion promotes cardiomyocyte proliferation in neonatal mice [23]. Currently, it is unclear which specific signals and downstream pathways mediate neural-dependent regulation of cardiomyocyte proliferation during these developmental periods.

## Studying neural-dependent heart growth in *Ciona robusta*

The tunicate *Ciona robusta* represents a powerful model to explore the role of neural inputs in heart morphogenesis or growth. Tunicates are chordates and represent the closest extant invertebrate taxa to the vertebrates. This evolutionary relationship is most evident in the tunicate larval tail which exhibits two key chordate traits, a notochord and dorsal nerve cord. However, both of these features are lost during metamorphosis as the larval tail is resorbed. In the resulting juveniles, the central nervous system is restructured and rudiments contained in the larval head/trunk region undergo morphogenesis to form adult organs, including the heart. Despite this close evolutionary relationship, tunicate embryos, including those of *C. robusta*, have extremely low cell numbers and highly stereotyped cell lineages facilitating high-resolution analysis of cell fate specification and morphogenesis. This cellular simplicity has been leveraged to gain profound insights into early stages of *C. robusta* heart development, along with comprehensive mapping of the larval nervous system and in-depth analysis of the neural circuits driving larval swimming behaviors [83–85]. Efforts to analyze neural and cardiac development during *C. robusta* embryogenesis have also been facilitated by the small size of the *C. robusta* genome and an associated lack of genetic redundancy. Recently, *C. robusta*'s extreme cellular and genetic simplicity were exploited to permit comprehensive scRNA sequencing of embryonic stages and reconstruction of the transcriptional trajectories for every cell lineage [86]. However, studies of *C. robusta* development have largely focused on embryogenesis. Thus, post-larval development, including morphogenesis and subsequent growth of the adult heart or nervous system, remain poorly characterized.

## Cellular composition of the *C. robusta* heart

The *C. robusta* heart is primarily constructed from two single-cell layers, the myocardium and pericardium (Fig 1A) [87]. The myocardium is a V-shaped tube consisting of myoepithelial cells capable of generating peristaltic bi-directional contractions. Embedded within one side of the myocardium is the undifferentiated line (UL). The UL is a tightly packed group of small round cells arranged into rows spanning the entirety of the myocardial tube [88]. The classification of these cells as undifferentiated is based solely on appearance and their function has not been previously characterized. The

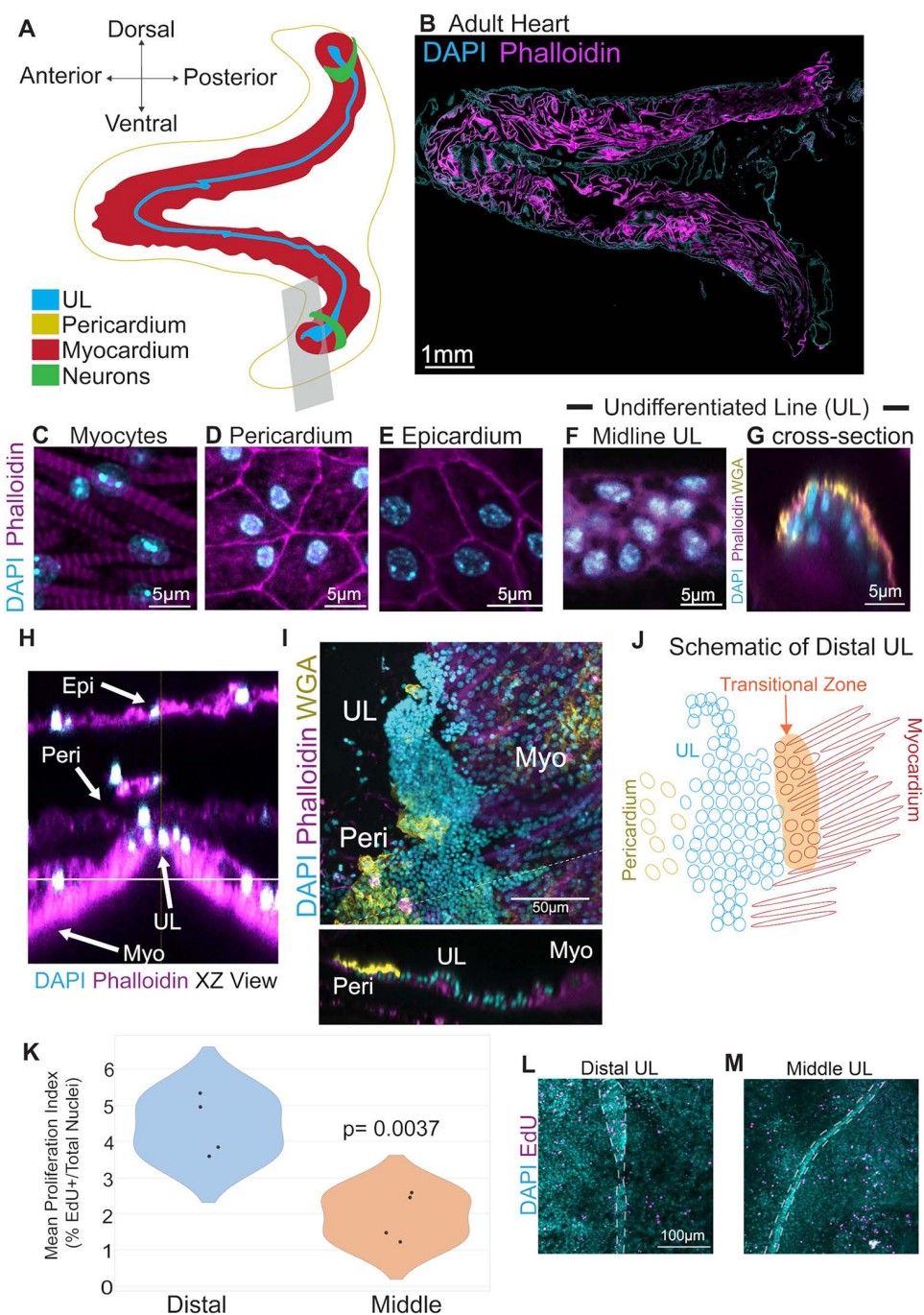

**Fig 1. Microscopic anatomy of the adult *Ciona robusta* heart. (A)** Diagram depicting gross anatomy of the adult *C. robusta* heart. **(B)** Confocal micrograph of an adult *C. robusta* heart. **(C–F)** Cardiac cell types as labeled. **(G)** Cross-section of the adult midline UL. **(H)** Cross-section (X-Z) showing the relative position of different cardiac layers (Epi = epicardial, Peri = pericardial, UL, undifferentiated line, Myo = myocardial). **(I)** Distal end of an adult heart showing expanded UL along with adjacent pericardium and myocardium. Panel below shows a cross-section of **I**. **(J)** Diagram of the distal portion of the heart (**I**) highlighting a presumptive transitional zone, see text. In panels **A–I**, cyan represents DAPI staining, magenta represents phalloidin staining and yellow represents WGA staining. **(K)** Graph displaying mean proliferation index (% dividing cells) in distal and middle regions of the heart as assayed by 24-hour pulses of EdU, $N = 8$ adult hearts. We first tested for normality using the Shapiro–Wilk Test ($W = 0.8938$, $p = 0.4010$), and then proceeded with a Student $t$ test. The data underlying this Figure panel can be found in S1 Data. **(L–M)** Representative images of proliferation in the distal (**L**) and middle (**M**) portions of the UL (outlined by a white dashed line) as assayed by a 6-hour pulse of EdU (magenta).

pericardium forms a coelom around the myocardial tube and is connected to the myocardium on the opposite side of the heart from the UL by a structure termed the raphe. Superficial to the pericardium is an epithelial lining termed the "epicardium" which surrounds the stomach, intestine, gonad, and heart [89]. Our current understanding of *C. robusta* heart anatomy is largely based on classic histological studies that have not been updated in over 50 years.

**Intrinsic innervation of the *C. robusta* heart**

As initially characterized by Kriebel and colleagues [90–93], tunicates have pacemakers at either end of the myocardial tube that regulate bi-directional pumping of the heart [92–94]. Although some classic studies mention extrinsic neural innervation of the heart, more recent analyses have not detected these extrinsic neurons [95–97]. A recent study identified rings of neural-like cells that are integrated within the distal ends of the myocardium (green rings, Fig 1A) [98]. These cells were visualized using a stable transgenic line in which a *Prohormone Convertase 2* (*PC2*) regulatory element is used to drive expression of a Kaede fluorescent reporter. *PC2* plays a highly conserved role in neuropeptide processing and thus expression of *PC2* reporter suggests that these distal neural rings are peptidergic. As pointed out in this previous study, the position and morphology of the distal plexuses are reminiscent of the intrinsic sinoatrial and atrioventricular plexuses present in vertebrate hearts. The authors of this study proposed that these distal plexuses function as pacemakers. However, the function and development of these distal peptidergic rings has not been characterized and no connections to extrinsic neurons originating outside of the heart were recorded.

In this paper, we delineate the microanatomical composition of the *C. robusta* heart and characterize a line of presumptive cardiac progenitors that appear to drive proliferative heart growth. We have also begun to explore the signaling pathways that regulate heart growth, including a key role for canonical Wnt (cWnt) signaling in suppressing progenitor proliferation. Strikingly, we identified a set of neural-like cells that are closely associated with cardiac progenitors. These cardiac neural-like cells appear to be innervated by extensions from the CNS during early juvenile stages. Single cell-RNA sequencing of the adult heart indicated that these cardiac neural-like cells express the receptor for tachykinin-family neuropeptides (TACR). To begin assessing the role of tachykinin signaling in these cells, we applied a pharmacological inhibitor to the adult heart. Surprisingly, this inhibitor had no impact on heart rate but rather suppressed heart cell proliferation. Transient inhibition of tachykinin signaling in juveniles specifically decreased proliferation in the distal progenitor lineages adjacent to intrinsic neurons. Targeted CRISPR knockdown of either *TACR* in cardiac lineage cells or *tachykinin* in extrinsic neurons led to nearly complete abrogation of the myocardium. Taken together, these results indicate that tachykinin-dependent stimulation of cardiac neural-like cells is required for cardiomyocyte progenitor proliferation.

## Results

**Cellular composition and growth of the adult *C. robusta* heart**

To lay the groundwork for exploring tunicate heart growth, we generated a high-resolution microanatomical whole-mount confocal map of the adult *C. robusta* heart (Fig 1B). Each of the four cardiac cell types (UL, myocardium, pericardium, and epicardium) were readily distinguished based on their morphology and position (Fig 1C–1H). Myocardial cells contain highly organized myofibers (Fig 1C). The pericardium is immediately superficial to the myocardium and consists of flattened, roughly pentagonal cells with peripheral nuclei (Fig 1D and 1H). The epicardial cells, distinguishable by their more irregular shape and central nuclei, form a thin layer superficial to the pericardium (Fig 1E and 1H).

We were particularly interested in the microanatomy of the UL. In the mid-section of the heart, the UL consisted of 3–5 irregular rows of compact cells (Fig 1F). In cross-section, these rows form a semi-cylinder that is attached to myocardial cells on either side (Fig 1G–1H). As observed previously by Millar, the UL expands at both the dorsal "visceral" and ventral "hypobranchial" ends of the heart to form larger clusters (Fig 1I and 1J) [88]. Extensive folding made it extremely difficult to visualize the morphology of these distal UL clusters in intact hearts. To overcome this obstacle, distal regions of the heart were removed and dissected to form a flat sheet (Fig 1I and 1J). The observed distal UL clusters were highly

variable in size and morphology. As seen in the mid-section of the heart (Fig 1G), the central cells of the distal UL clusters form a raised semi-cylinder (Fig 1I, cross-section). In some cases, the distal UL appeared to display a density gradient, with an extremely dense region at the distal-tip and a wider, less dense proximal region (see Fig 1I). In this wider proximal region, the UL cluster appeared to be flanked by different cell types on each side (Fig 1I and 1J). One side of the UL cluster appears to merge into a field of pericardial cells, as distinguished by their flat, hexagonal shape and enriched WGA staining. On the opposite side, streams of distal UL cells appear to penetrate into regions containing mature myocardial cells, as distinguished by their spindle-like morphology and dense bands of actin. There appears to be a morphological gradient between the distal UL cluster and mature pericardium as tightly packed UL cells with highly condensed nuclei transition into sparsely distributed pericardial cells with larger nuclei (Fig 1I and 1J). There also appears to be a morphological gradient associated with myocardial differentiation as penetrating streams of UL cells give way to a zone of less densely packed cells with larger nuclei and lower levels of actin enrichment (Fig 1H–1J). These transitional cells may constitute a population of myocardial precursors.

We next investigated the pattern of growth in adult *C. robusta* hearts. As posited by Millar, distal UL clusters may serve as proliferative growth zones for heart tube elongation [88]. To test this hypothesis, we assayed gross proliferation rates in distal versus middle regions of the myocardial tube (including both myocardial and UL cells) using 24-hour EdU pulses. As predicted by the distal growth zone hypothesis, the distal myocardium displayed a significantly higher proportion of mitotic cells in comparison to middle regions (Fig 1K). To further characterize the location of mitotically active cells, we used a short 6-hour EdU pulse assay followed by a 24-hour chase (Fig 1L and 1M). In distal regions, EdU+ cells were largely observed within or bordering the distal UL cluster (Fig 1L). By contrast, EdU+ cells were rarely detected in the midline UL (Fig 1M). Additionally, scattered EdU+ cells were detected throughout the myocardium. The proliferative distal UL border cells may represent a transitional zone, in which myocardial or pericardial precursors undergo proliferation prior to differentiation (as illustrated in Fig 1J). Further analysis of adult heart growth was limited by the convoluted morphology of the distal cluster and difficulties in obtaining intact, dissected samples. To overcome these challenges, we began to examine heart growth during early juvenile stages.

## Cellular composition of the juvenile heart

Due to extremely low cell numbers, distinct cellular components of the juvenile heart can be readily distinguished (Fig 2A and 2B). The heart first starts beating three days after fertilization (D3), approximately two days after larval settlement. Although early-stage juvenile hearts (D3–D6) are composed of less than 100 cells, they are fundamentally similar in structure to the adult organ (Fig 1A and 1B). An inner, cylindrical myocardial tube is surrounded by a roughly spherical pericardial layer composed of ~70 flattened epithelial cells. The myocardial tube is composed of ~10–24 cardiomyocytes attached along the posterior side to a UL composed of only ~8–14 cells. As in the adult, the UL consists of a narrow medial line abutted by expanded distal clusters. However, in the juvenile, the medial line consists of a single row and the distal clusters are composed of only two or three cells. Each medial UL cell appears to be associated with a ring-like pair of spindle-shaped myocardial cells (Fig 2A and 2B). As juveniles age, the UL gradually lengthens (Fig 2C′–2C‴).

## Juvenile heart growth

To begin investigating juvenile heart growth, we quantified UL and myocyte numbers during early stages (D3–D15). UL cells and myocytes were readily identified based on their morphology and position (Fig 2A–2C). Under optimized growth conditions, our analysis indicated that an average of 1–2 midline UL cells and 4–5 myocardial cells are produced each day (S1 Fig). Notably, while the number of midline UL cells increased, distal cell numbers remained fairly constant. Distal clusters contained 2 or 3 cells except for occasional instances when there appeared to be only one isolated distal cell. We also observed occasional gaps in the otherwise continuous and tightly packed midline UL nuclei. Additionally, on examining these images we noted the occasional presence of cells located between the UL, which is on the posterior side of

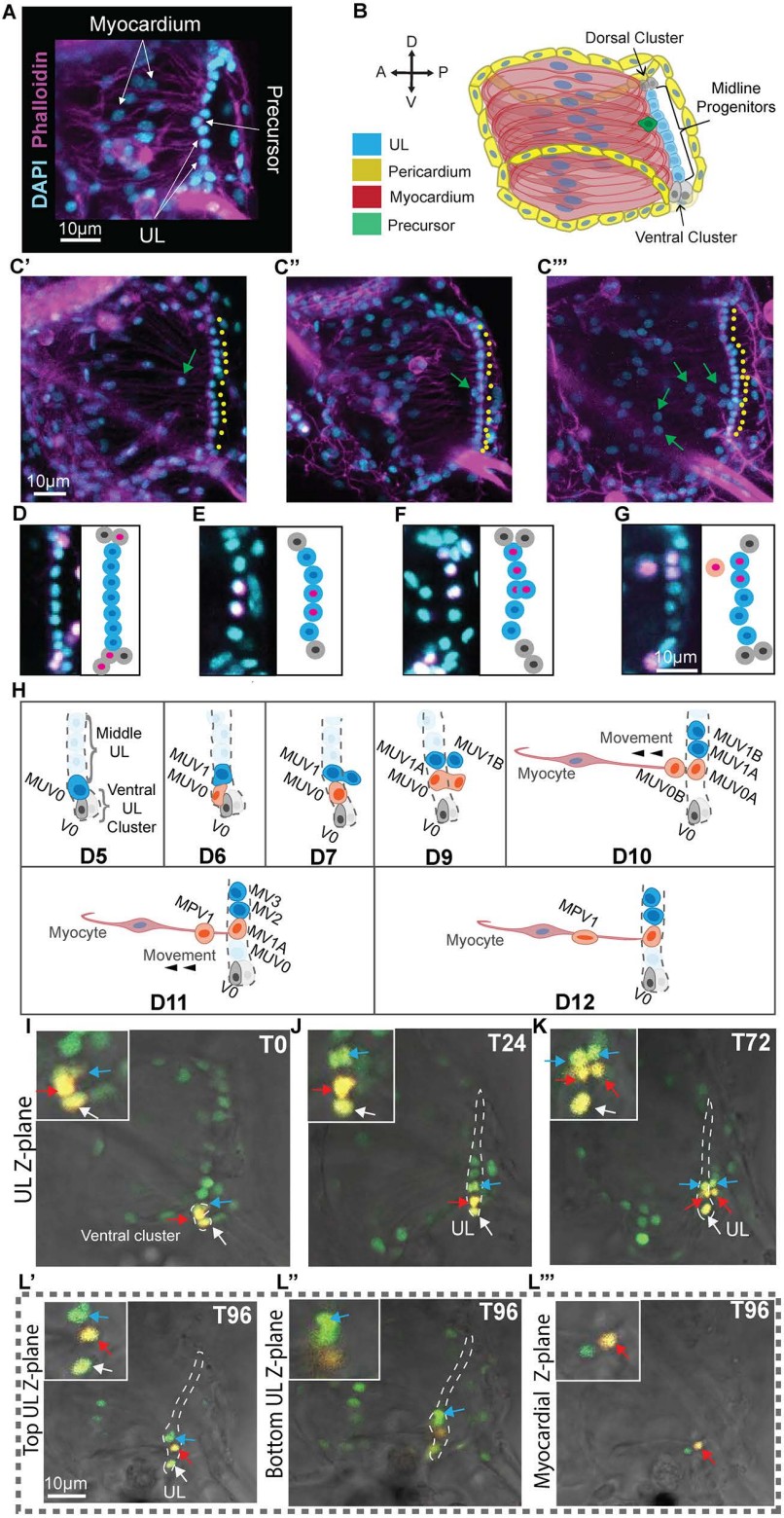

**Fig 2. Juvenile heart anatomy and cell division dynamics. (A)** D5 juvenile heart. **(B)** Schematic of **(A)**. **(C)** Representative images of D8 **(C′)**, D11 **(C″)**, and D15 **(C‴)** juvenile hearts. Yellow dots highlight midline UL nuclei **(C′**, 13 midline UL; **C″**, 15 midline UL; **C‴**, 18 midline UL). **(D–G)** D5 juvenile

hearts showing distinct EdU⁺ clones containing at least one UL as described in the text along with schematics shaded according to the model in panel **H**. **(H)** Model for UL and myocardial precursor cell division in juvenile hearts, based on data from D5-D12 juveniles in which photoconversion of Kaede-labeled ventral UL cells was used to track individual progenitors as represented for one sample in panels **I–L**. At the start of the model (first panel, Day 5) asymmetric division of a distal UL cell in the ventral cluster has produced one distal UL daughter (gray, V0) and one midline UL daughter (dark blue, MUV0). Subsequently (second panel, D6) symmetric division of the newly produced midline UL (MUV0) produces two midline daughters (MUV0 shown in red and MUV1 shown in blue). Over the next few days, both midline daughters divide. One (MUV1, blue) divides symmetrically to produce two daughters that remain in the midline (MUV1A and B, blue). The other (MUV0, red) divides asymmetrically to produce one midline UL daughter (MUV0A, red) and a myocardial precursor that leaves the midline (MUV0B, red). It is not clear from our data whether myocardial precursors are derived from the posterior or anterior daughter during these asymmetric divisions. According to this model, each myocardial precursor migrates anteriorly as it differentiates, eventually contributing to the myocardial tube (D10–Dd12 in the model). **(I–L)** Tracking of UL cells near the ventral cluster that were labeled by Kaede conversion (yellow). Cells were photoconverted on D5 (T0) and imaged daily over four days (T24–T96). See text for details of division patterns and cell movements of labeled cells (as illustrated in panel **H**). Different colored arrows track three individual photo-converted cells over 96 hours after conversion, as indicated by the time point (T0-T96). Magnified regions shown in the upper left corner of each image. The red, blue and white arrows correspond to the red, blue, and gray cells in the model shown in panel **H**. Green cells are unconverted Kaede while photoconverted cells are yellow. **(L′–L‴)** For the T96 time point, three different sub-stacks of the same Z-stack are shown to clarify the relative locations of daughter cells from the three original photoconverted cells across three different focal planes as labeled. Two of the focal planes (Top UL and Bottom UL) contain other labeled or unlabeled UL cells while the third focal plane (myocardial) is distinct from the UL.

the heart, and the myocardial nuclei, which form two bands in a more anterior region of the heart (green cell in 2B, green arrows in 2C). Based on their location along with the lack of the elongated nucleus or association with actin-enriched fibers characteristic of differentiated myocardial cells, we hypothesize that these cells represent myocardial precursors.

## Clonal analysis of the undifferentiated line

To further explore the potential contribution of the UL to myocardial growth, we performed a series of EdU pulse/chase experiments. We first established that a 30-min EdU pulse could be reliably used to label single UL cells for clonal analysis. In particular, we found that when D3 juveniles were exposed to a 30-minute pulse of EdU, the majority (87/110) exhibited no EdU⁺ UL nuclei while all but one of the remaining samples exhibited a single EdU⁺ UL nucleus (22/110). In the one sample with two EdU⁺ UL cells, labeled nuclei were widely separated, one in the dorsal distal cluster and the other in the ventral distal cluster. Because the pericardium is highly proliferative, we sometimes observed adjacent labeled UL and pericardial cells, but we did not observe adjacent labeling of UL and myocardial precursors. Further analysis of these data indicated that distal UL cells have a substantially higher proliferative index (6.2%) than midline UL cells (1.7%). To initiate clonal analysis, we subjected D3 juveniles to a 30-min EdU pulse followed by a 72-hour chase. We then assessed the composition of presumed clones across 14 independent samples containing at least one EdU+ UL cell. These labeled clusters often consisted of 1–4 EdU+ cells indicating that up to two divisions had occurred during this 72-hour period. On the distal end, we observed clones containing paired distal and midline UL cells (Fig 2D). In the midline, we often observed pairs of adjacent midline cells (Fig 2E). In rare instances, we observed what appear to be newly divided midline cells in which one daughter was located posterior to the UL (Fig 2F). Additionally, we observed triplet clones consisting of two adjacent midline UL cells associated with a presumptive myocardial precursor (Fig 2G). Based on these observations, we generated a model for heart growth (Fig 2H). According to this model, distal UL clusters function as self-renewing progenitor populations, dividing asymmetrically to produce midline cells while maintaining a steady population of distal progenitors. Each resulting midline cell either divides symmetrically to further increase UL length or asymmetrically to generate a myocardial precursor. The newly produced precursors then detach from the UL and migrate anteriorly along existing myofibrils as they differentiate.

To more definitely test the hypothetical contribution of the UL to myocardial growth, we traced individual midline UL cell lineages using a photoconvertible Kaede reporter. The *Mesp* regulatory element [99] was used to drive expression of a green◊red photoconvertible Kaede protein [100] in heart cell nuclei in transgenic juveniles (*Mesp>Kaede::NLS*). In each sample, either a single midline nucleus or a cluster of nuclei near the ventral or distal end of the UL was photoconverted and juveniles were then cultured separately. Photoconverted lineages in cultured juveniles were imaged every 24 hours

for up to five days. As predicted by the low proliferative index observed in our EdU pulse experiments (see above), UL cell divisions were rarely observed. Despite this challenge, we were able to document divisions of midline UL cells that appeared to be either symmetric (Figs 2I–2L and S2A) or asymmetric (Figs 2I–2L, S2B, and S2C) as predicted by the EdU clonal data. In these studies, we defined division symmetry based on the relative position of the resulting daughter cells. If both daughter cells remained aligned with neighboring UL cells (and in the same focal plane), we considered this as representing a symmetric division. On the other hand, if one daughter was aligned with neighboring UL cells while the other daughter was no longer aligned (having been displaced either anteriorly or onto a different Z-plane), we considered this as representing an asymmetric division. Critically, we were able to document a number of "asymmetric" midline UL divisions (Figs 2I–2L, S2B, and S2C). We also observed a single photoconverted midline UL cell that appeared to represent a myocardial precursor, migrating anteriorly over a four-day period (S2D Fig). In another sample (S2E Fig), we labeled a single presumed myocardial precursor that appeared to differentiate as indicated by shifts in nuclear morphology, eventually displaying a narrow, ovoid nucleus characteristic of differentiated myocardial cells. In one of our samples, we were able to observe the division of two labeled, adjacent midline UL cells close to the ventral end of the UL (Fig 2I–2L, blue versus red arrows). As observed in our EdU clonal assay (Fig 2F), both of these cells appeared to divide along the anterior/posterior axis producing one daughter that remained in the UL and one daughter that was displaced posteriorly in relation to the UL (see arrows in Fig 2K). Intriguingly, one of these midline UL cells appears to divide "symmetrically" producing two daughters that remained associated with the UL (blue arrows in Fig 2I–2L, blue-shaded MUV1 lineage in Fig 2H) while the neighboring cell appeared to divide "asymmetrically" producing one daughter that remained associated with the UL while the other daughter appeared to migrate anteriorly and along the Z-axis, ending up on a different confocal plane (red arrows in Fig 2I–2L, red-shaded MUV0 lineage in Fig 2H). Based on this data, we have further refined our model of UL division and heart growth (Fig 2H). In particular, we propose that once a new midline UL cell is generated following asymmetric division of a distal UL, it divides to produce midline UL daughters that exhibit distinct division patterns. One daughter subsequently divides symmetrically while the other daughter divides asymmetrically to produce a myocardial precursor. Although the details of this model may shift as more data is collected, our data supports the following premise: The UL constitutes a cardiac progenitor population that plays a central role in myocardial growth. In accordance with this premise, we will subsequently refer to UL cells as cardiomyocyte progenitors.

### Transcriptional profiling of *C. robusta* heart cell types

To delineate the transcription profile of the cardiomyocyte progenitors and other heart cell types we performed scRNA-seq of two adult heart samples (Fig 3). Protocols were optimized to ensure dissection resulted in intact hearts that were subsequently dissociated into pools of viable cells that represented comprehensive and distinguishable cell types. To ensure high viability during encapsulation, we deployed a protocol developed by the Klein Lab [101]. As seen in relation to blood cells, the use of cell encapsulation solutions that better match the osmotic environment of cells in marine organisms produced a substantial increase in viability for dissociated *C. robusta* heart cells [101]. Bioinformatic analysis was performed as described (see Methods) to identify distinct cell states. Because dissected hearts contain blood cells, we aligned our data against existing scRNA-seq data from the Klein Lab [102] and found that many of our clusters closely matched blood cell states (gray in Fig 3A), leaving us with eight presumptive heart cell clusters (Fig 3A). Nearly all these clusters were distinguished by highly enriched expression for a number of transcription factors (Fig 3B) along with other marker genes (Fig 3C–3F). Interestingly, the central cluster (Cluster 1, Fig 3A) was not as transcriptionally distinct in regards to both of these criteria. The lack of distinct expression for Cluster 1 was also evident when we ran a Pearson's correlation coefficient analysis to assess cluster similarity based on global gene expression (S3 Fig). In particular, this analysis indicated that Cluster 1 was most similar to Cluster 7.

We next used marker gene expression to provisionally assign each of these eight clusters to one of the distinct cell types identified through our microanatomical analysis (Fig 3D). Through this approach, we identified two presumptive

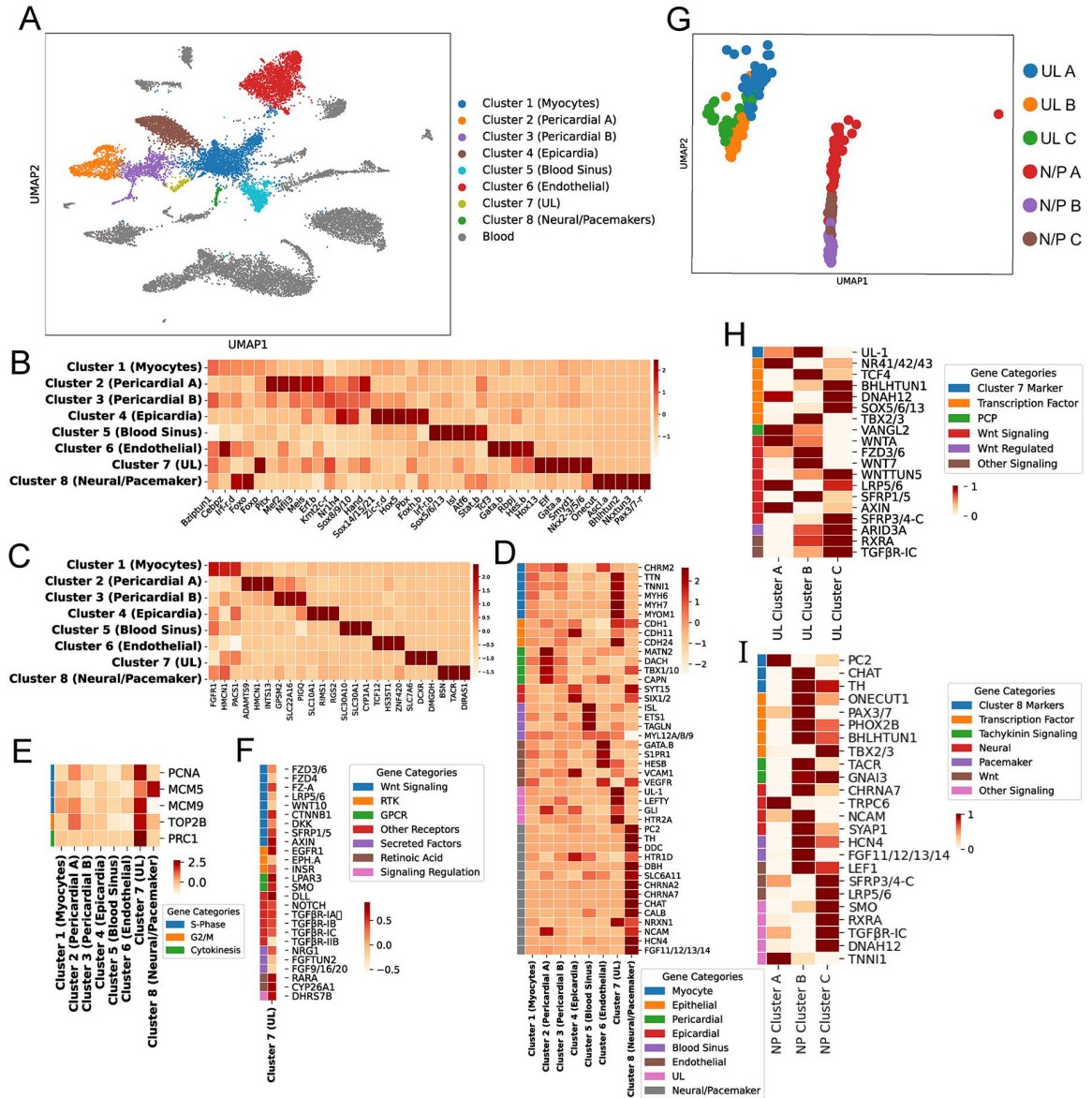

**Fig 3. Transcriptomic analysis of adult heart scRNA-seq data. (A)** UMAP of scRNA-seq data. Cardiac clusters colored according to Louvain clustering, presumptive blood cell clusters in gray. **(B)** Heatmap of transcription factor expression. **(C)** Heatmap displaying three of the most highly enriched marker genes for each cluster. **(D)** Heatmap of additional marker genes associated with each cluster. **(E)** Heatmap of marker genes for cell proliferation. **(F)** Heatmap of signaling pathway genes in Cluster 7. Z-score was derived from comparing all 8 heart clusters, but only Cluster 7 is shown. **(G)** Louvain sub-clustering analysis of Cluster 7 (UL A, B, and C) and Cluster 8 (Neural/Pacemaker, N/P A, B, and C). **(H–I)** Heatmaps comparing expression for genes of interest across each UL sub-cluster **(H)** or Neural/Pacemaker sub-cluster **(I)**. Data for this figure can be accessed at the NCBI Sequence Read Archive, BioProject accession PRJNA1420413. SRR37160362 corresponds to heart sample 1 FASTQ files, and SRR37288072 corresponds to heart sample 2 FASTQ files.

pericardial clusters, Pericardial-A (Cluster 2) and Pericardial-B (Cluster 3). Previous studies have indicated that the *C. robusta* pericardium arises from the second heart precursor (SHP) lineage [84], and both of these clusters express known SHP markers including *Matrilin* (*MATN*)*1/3/4* and the transcription factor *Dachshund* (*DACH*) (Fig 3D; see S1 Table for list of genes analyzed for this figure panel) [103]. The strong correlation between these two clusters (S3 Fig) supports

their co-assignment as pericardial sub-types. Cluster 1 was difficult to assign due to the lack of unique markers. However, we tentatively designated this cluster as representing differentiated myoepithelial cells due to weakly enriched expression for numerous orthologs to muscle-related genes including *Muscarinic Cholinergic Receptor 2* (*CHRM2*), *Titin* (*TTN*), *troponin-like 1* (*TNNL1*), *Myosin Heavy Chain 6* (*MYH6*), *myosin heavy chain 7* (*MYH7*), and *Myomesin-1* (*MYOM1*) (Fig 3D). We designated Cluster 4 as representing epicardial cells based primarily on reporter analysis for the marker gene synaptotagmin-15 (*SYT15*) (S4A Fig). Additionally, this cluster displays enriched expression of the ortholog to *Sine Oculis homeobox 1/2* (*Six1/2*), a transcription factor that is required for mammalian epicardial progenitor development [104]. We hypothesize that Cluster 5 represents cells lining the blood sinuses adjacent to the heart, which may be equivalent to the vertebrate outflow tract [105]. This cluster displays enriched expression of orthologs to Islet (*Isl*) and *E26 transformation-specific or Erythroblast Transformation Specific* (*ETS1*), which are known transcriptional markers for *C. robusta* blood sinus cells [106]. In addition, both *Isl* and *ETS1* play conserved roles in vertebrate outflow tract specification [107,108]. We also noted that Cluster 5 displays specific, highly enriched expression of the orthologs to *transgelin* (*TAGLN*), an established cardiac outflow tract smooth muscle marker [109] and the smooth muscle marker *myosin light chain* (*MYL*)-*12A/12B/L9* [110]. Our data indicate that Cluster 6 may represent a population of endothelial-like cells. Although endothelial cells are considered to be a vertebrate-specific innovation [111], it is possible that a precursor cell-type exists in *C. robusta*. In line with this hypothesis, Cluster 6 displays enriched expression of the ortholog to *VEGFR*, a key marker of vertebrate endothelial cells [112]. Cluster 6 also displays enriched expression *Gata.b*, the *C. robusta* ortholog of *GATA-2*, which is necessary for proper endothelial development in vertebrate models [113,114]. Additionally, Cluster 6 displays expression of orthologs to the endothelial and smooth muscle marker *sphingosine-1-phosphate receptor 1* (*S1PR1*) along with *hairy and enhancer of split B* (*HES-B*) a transcription factor that is required for endothelial blood vessel-fate specification in mice [115].

We were particularly interested in identifying which of the presumptive heart cell clusters represented the cardiomyocyte progenitors (UL). Cluster 7 displays highly enriched expression of transcription factors associated with early embryonic cardiac lineage specification in *C. robusta* and other organisms, including the sole *C. robusta* orthologs to *GATA4/5/6* and *Nkx2.5* (Fig 3B). These expression patterns may reflect re-initiation of the embryonic cardiogenic specification program during post-larval cardiogenesis. Cluster 7 also displays expression of the ortholog to *SET and MYND domain containing 1* (*SMYD1*) which is necessary for embryonic heart proliferation and morphogenesis in mice [116–118]. To further explore whether Cluster 7 represents the heart progenitor population, we examined the relative expression of five genes associated with proliferation. This analysis revealed strong, and relatively specific, enrichment for four of these proliferative marker genes in Cluster 7 (Fig 3E).

To more definitely test the hypothesis that Cluster 7 represents the cardiomyocyte progenitor lineage (UL), we examined the in situ expression pattern for the most uniquely enriched gene in this cluster (Fig 3D, UL1 previously annotated as KH.C1.1154 or as NCBI gene entry XM_026835482). This gene encodes a tunicate-specific, uncharacterized protein that we have named *CrUL1*. In situ expression analysis in juvenile samples demonstrates that *CrUL1* is only expressed in the heart (S4B Fig). Critically, as predicted by our clustering analysis, *CrUL1* expression is strongly enriched in cardiomyocyte progenitors, including both midline and distal portions of the UL (S4B Fig). *CrUL1* is also weakly expressed in a bulging region on the opposite side of the heart, potentially associated with the raphe. Intriguingly, *CrUL1* mRNA is strictly localized to the anterior side of each progenitor cell nucleus. The CrUL1 protein is predicted to have a secretory signal peptide [119] and thus, anterior localization of *CrUL1* mRNA may facilitate unidirectional secretion into the ECM on this side. We also conducted in situ expression analysis for *Gata.a* (S4C Fig). As predicted by our adult heart scRNA-seq data, *Gata.a* is highly expressed in the UL in juvenile samples. However, we also observe *Gata.a* expression in the juvenile myocardial cells. The apparent loss of *Gata.a* expression in the presumed adult myocyte sc-RNA-seq cluster (Fig 3B) may reflect definitive myocyte differentiation during the juvenile to adult transition.

We noted that the expression of some TFs and markers appeared to be more enriched in specific sets of cells within the putative cardiomyocyte progenitor cluster (Cluster 7). To further explore these differential expression patterns, we

performed an additional Louvain clustering analysis on Cluster 7 alone leading to the identification of three sub-clusters (UL-A, B, and C, Fig 3G and 3H). UL-A displayed enriched expression of the ortholog to transcription factor *nuclear receptor subfamily 4 group A* (*NR4A*), which regulates the cardiac response to neurohormonal signaling in mice [120]. UL-B is distinguished by high levels of expression of the sole *C. robusta* ortholog to *Tbx2* and *Tbx3*, transcription factors associated with specification of the atrio-ventricular canal and associated cardiac conduction cells in mice and zebrafish [70,121]. We also noted that the UL marker gene *CrUL1* is primarily enriched in UL-B. UL-C is distinguished by high levels of expression for orthologs to *SRY-Box Transcription Factor 5/6/13* (*Sox5/6/13*), *basic helix-loop-helix tunicate 1* (*Bhlhtun1*), and *dynein axonemal heavy chain 12* (*DNAH12*). Intriguingly, the latter two genes are both established markers for the *C. robusta* inner atrial siphon muscle precursors, a sister lineage to the heart precursors and *Bhlhtun1* has been shown to promote self-renewal in these siphon muscle precursors [122].

## Expression of signaling pathway genes in cardiomyocyte progenitors

We next assessed the expression of orthologs to receptors or downstream target genes from a range of developmental signaling pathways in the presumed cardiomyocyte progenitor scRNA-seq cluster (Cluster 7, Fig 3F). This cluster displayed enriched expression for a number of orthologs to signaling receptor genes including *Epidermal growth factor receptor-a* (*Egfr.a*), *Ephrin-a* (*Eph.a*), *Notch*, *Insulin receptor* (*InsR*), *Transforming growth factor* (*Tgf-β receptor-Ia, -Ib, -Ic*), and *Retinoic Acid Receptor alpha* (*Rar.a*). Orthologs to *cytochrome P450 superfamily* (*CYP26A1*) and *dehydrogenase/reductase 7B* (*DHRS7B*) that are involved in feedback inhibition of RA signaling were also highly expressed in the UL cluster (Cluster 7). Notably, the presumed progenitor UL cluster displayed expression for numerous orthologs to genes associated with cWnt signaling. These included Wnt receptors such as *Frizzled* (*Fzd-3/6/7*), *Fz4* and *Fz-a*, the co-receptor *LDL Receptor Related Protein* (*LRP-5/6*), along with *Wnt10A* and *β-Catenin-1* (*CTNNB1*). UL clusters also displayed enriched expression of orthologs to three genes that are characteristically up-regulated in response to cWnt signaling to mediate auto-negative feedback, *Dickkopf* (*DKK*), *secreted frizzled-related protein 1* (*sFRP1*), and *Axin* [122–124]. As various cWnt signaling components were highly expressed in the UL, we also examined the expression of orthologs to cWnt pathway genes within the UL sub-clusters (Fig 3I). Based on these findings, we began to assess the contribution of cWnt signaling to cardiac progenitor proliferation using pharmacological assays.

## Wnt signaling suppresses cardiomyocyte progenitor division

Based on previously characterized roles for cWnt signaling in retention of proliferative stem cell populations [125], we hypothesized that this pathway promotes *C. robusta* cardiomyocyte progenitor proliferation. To test this hypothesis, we treated juveniles with IWR-1-endo (IWR), a cell-permeable molecule that stabilizes Axin and thus promotes β-catenin degradation to disrupt cWnt signaling [124]. We then assayed the impact on progenitor proliferation using a 30-min EdU pulse assay. Strikingly, application of IWR led to a significant increase in the frequency of cardiomyocyte progenitor proliferation (Fig 4A and 4B). As seen in our clonal analysis, the majority of control juvenile hearts displayed no EdU+ progenitors and nearly all the remaining control hearts displayed one EdU+ progenitor. In contrast, following a 24-hour treatment with IWR, the majority of juvenile hearts displayed two or more EdU+ progenitors. A robust and significant increase in the incidence of EdU+ progenitors was seen in both midline and distal UL cells (Fig 4C and 4D). These results contradict our initial hypothesis, indicating that cWnt suppresses cardiomyocyte progenitor proliferation. To determine whether IWR also impacts the rate of cardiomyocyte progenitor division, we conducted EdU pulse/chase experiments. This analysis indicated that division rates were similar in control and IWR-treated cardiomyocyte progenitors (approximately one division per day, Fig 4E). Thus, cWnt signaling appears to specifically decrease the frequency of cardiomyocyte progenitor division and not the ensuing rate of division. To further test this hypothesis, we applied two cWnt agonists with distinct mechanisms of action, 6-Bromoindirubin-3′-oxime (BIO) [126] which blocks GSK3 activity, and SKL2001 which blocks the interaction between Axin and β-catenin [127]. Although we extended the EdU pulse length to 4 hours, labeling of progenitors

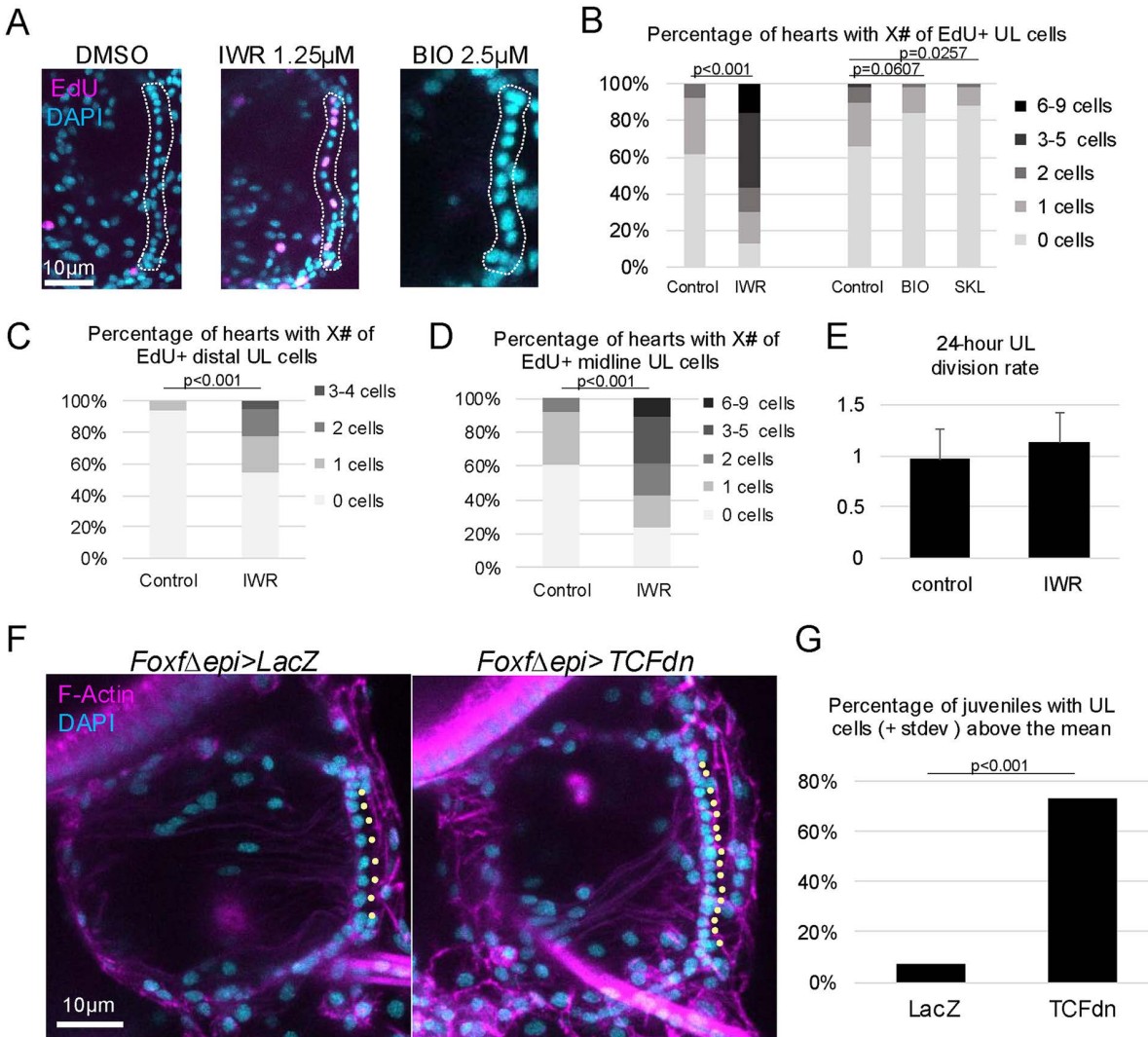

**Fig 4. Inhibition of the canonical Wnt pathway leads to an increase in the frequency of cardiomyocyte progenitor division. (A)** Representative images showing the incidence of EdU+ UL nuclei (magenta) in D7 juveniles treated with DMSO, the Wnt antagonist IWR-1-endo, or the Wnt agonist BIO. Dotted lines demarcate the UL of each heart. Cyan represents DAPI staining. **(B–D)** Graphs showing the % of hearts displaying specific numbers of all EdU+ UL cells **(B)**, EdU+ distal UL cells **(C)**, or EdU+ midline UL cells **(D)** in control and experimental samples as indicated. **(E)** Graph displaying mean UL division rates. To calculate these rates, juveniles were pulsed for 30 min with EdU followed by a chase period of 36 hours and assessed for the number of EdU+ nuclei. The average number of EdU+ nuclei in these chase experiments was then divided by the average number of EdU+ nuclei in the pulse-only experiments, then divided by 1.5 to get a per 24-hour value. A value of 1 indicates one division per day. 0 values were not included. **(F)** Representative images of hearts from D6 juveniles electroporated with the control construct *FoxfΔepi>LacZ* (left) or experimental construct *FoxfΔepi>TCFdn* (right). **(G)** Graph depicting the percent of juvenile hearts with substantially increased numbers of midline UL cells (greater than one standard deviation above the mean in controls). See Methods for statistical analysis. *T*-tests performed for **B**, **C**, and **D**. The data underlying the graphs in this Figure can be found in S1 Data.

remained infrequent in control juvenile hearts (Fig 4B). Despite this low baseline, we were still able to observe a reduction in the number of juvenile hearts displaying EdU+ progenitors in both treatment conditions and, in the case of the SKL2001-treated sample, this reduction was significant (Fig 4A and 4B). To alleviate concerns regarding the specificity of these pharmacological treatments, we used a targeted transgenic approach to disrupt the cWnt pathway. In these experiments, we deployed a characterized *Foxf* enhancer element [128] to drive a c-terminal truncated form of TCF specifically

in embryonic heart lineage cells (*FoxfΔepi>TCFdn*). *FoxF* enhancer-driven expression begins at the tailbud stage, well before heart morphogenesis and growth which occur after metamorphosis. Because the c-terminus of TCF mediates interaction with β-catenin, truncated TCF functions in a dominant negative fashion, suppressing cWnt-dependent gene expression [123,129]. Previous studies indicated that disruption of Wnt signaling using this construct did not interfere with cardiomyocyte specification [130]. Comparisons between hearts in transgenic *FoxfΔepi>TCFdn* juveniles versus control juveniles (*FoxfΔepi>LacZ*) revealed that transgenic disruption of the cWnt pathway in the heart lineage from an early time point consistently and significantly increased heart size along with the number of cardiomyocyte progenitors (Fig 4F and 4G). Quantification confirmed that there was a robust and significant increase in midline UL numbers in the experimental samples. Strikingly, despite this substantial increase in midline UL numbers, distal UL clusters did not appear to be expanded. Furthermore, there were no detectable abnormalities in the structure of these enlarged hearts (Fig 4F). Taken together, these results strongly support a role for cWnt signaling in suppressing cardiomyocyte progenitor proliferation, potentially contributing to the low proliferative index observed in these cells during early stages of heart growth.

### Transcriptional profile of neuron-like cells in the *C. robusta* heart

We noted that one of the cardiac cell clusters in our scRNA-seq data set (Cluster 8) displayed enriched expression for a number of orthologs to neuron-associated transcription factors including *One Cut Homeobox 1* (*Onecut1*) and *Achaete-scute complex-like-a* (*Ascl.a*) (Fig 3B) [131–133]. Additionally, this cluster was distinguished by unique and highly enriched expression of the gene encoding the sole *C. robusta* ortholog for the *tachykinin receptor* (*TACR*) (Fig 3C).

To begin exploring the identity and function of the cells in this presumptive neural-like cardiac cluster, we examined the expression of orthologs to other neural markers, including genes encoding neurotransmitter and neuropeptide signaling components within Cluster 8 (Fig 3D). This analysis revealed highly enriched expression for a number of orthologs to well-established neuronal markers including *Neurexin-1* (*NRXN1*), and *Neural cell adhesion molecule* (*NCAM*) [134,135]. Cluster 8 also displayed highly enriched expression of orthologs to a number of genes involved in the synthesis and processing of dopamine including—*tyrosine hydroxylase* (*TH*), *dopa decarboxylase* (*DDC*), and *dopamine beta-hydroxylase* (*DBH*) which participates in converting dopamine into noradrenaline. We also noted that this cluster displayed enriched expression for orthologs to a number of genes encoding neurotransmitter receptors including *serotonin receptor subunit 5-hydroxytryptamine receptor 1D* (*HTR1D*) and *nicotinic alpha 2 subunits-2 and 7* (*CHRNA2* and *CHRNA7*) [136,137]. Additionally, Cluster 8 displayed enriched expression of an ortholog to *Solute carrier family 6 member 11* (*SLC6A11*), involved in uptake of the inhibitory neurotransmitter gamma-aminobutyric acid (GABA). Lastly, this neural-like cluster displayed enriched expression of the ortholog to *Choline acetyltransferase* (*ChAT*), which is necessary for the biosynthesis of acetylcholine. Thus, the gene expression profile for this cluster clearly indicates representation of neural-like cells associated with the heart.

Strikingly, Cluster 8 displayed highly enriched expression of the ortholog to *prohormone convertase 2* (*PC2,* Fig 3D), which is a neuropeptide enzyme necessary for cleaving tachykinin and other neuropeptides into functional forms. PC2 reporter signal has previously been shown to label neural plexuses that form rings around the distal ends of the *C. robusta* myocardial tube [96], leading us to hypothesize that Cluster 8 may contain cells from these cardiac neural rings. Intriguingly, Cluster 8 also displayed enriched expression of orthologs to vertebrate marker genes for intrinsic cardiac neurons including *ChAT*, *DBH*, and *Calbindin* (*CALB*). [137–139] along with marker genes for cardiac pacemakers including *hyperpolarization-activated cyclic nucleotide-gated potassium channel 4* (*HCN4*), *fibroblast growth factor 13* (*FGF13*), and *TH* [137,138,140]. Taken together, our analysis suggests that Cluster 8 may contain a variety of cell types including both intrinsic neurons and potential pacemaker cells that comprise the cardiac conduction system.

To determine if there are indeed transcriptionally distinct neural-like cell-types contained within Cluster 8, we performed an additional Louvain clustering analysis on this cluster alone leading to the identification of 3 preliminary sub-clusters (Fig 3G, note that these subclusters are not definitive and may each represent a number of distinct cell types with similar

expression patterns). We then examined the expression of a variety of orthologs for neural genes within these sub-clusters (Fig 3I). Neural/Pacemaker-A (NP-A) consisted of 54 cells that were distinguished by highly enriched expression of the orthologs to *PC2* and *Transient receptor potential cation channel subfamily C, member 6* (*TRPC6*), which is modulated by TACR signaling [141]. Neural/Pacemaker-B (NP-B) consisted of 50 cells that were distinguished by highly enriched expression of the *C. robusta* orthologs to *CHAT*, *PAX3*, and the *tachykinin receptor (TACR).* NP-B also displayed relatively high levels of expression for orthologs to the cardiac pacemaker markers *HCN4* and *FGF13*. Neural/Pacemaker-C (NP-C) consists of 26 cells that were distinguished by highly enriched expression of orthologs to the transcription factor *DNAH12* as well as *Tbx2/3*, the sole *C. robusta* ortholog to *TBX2* and *TBX3*, a pair of paralogous transcription factors that play a key role in development of the vertebrate cardiac conduction system [70,121]. NP-C also expressed relatively high levels of the ortholog to the pacemaker marker *HCN4* and was distinguished by enriched expression of orthologs for a number of genes associated with signaling pathways, including the RA, Wnt, hedgehog, and TGF-B pathways. Based on this analysis, we hypothesized that NP-A represents a population of intrinsic cardiac neurons while NP-B and NP-C represent mixed populations containing both intrinsic neural and pacemaker cell types. Note that this interpretation is highly provisional. As detailed below, we have conducted reporter analysis for a limited set of sub-cluster-specific marker genes. However, further studies will be required to accurately characterize cell types represented by this cluster.

## Profile of cardiac intrinsic innervation in adult *C. robusta* hearts

We next deployed stable transgenic lines to begin characterizing the location of presumptive neural/pacemaker cell types in the adult *C. robusta* heart (S5 Fig). Based on the enriched expression of *PC2* in sub-cluster NP-A (Fig 3I), we used a *PC2>Kaede* reporter to label peptidergic neurons potentially represented by this sub-cluster. As observed previously [96], this reporter labeled neural rings associated closely with the distal myocardium at both the dorsal (visceral) and ventral (hypobranchial) ends of the heart (S5B Fig). Based on the enriched expression of *ChAT* in sub-cluster NP-B (Fig 3I), we used a *VACHT>CFP* reporter to label cholinergic neural-like cells potentially represented by this sub-cluster. *ChAT* (choline acetyltransferase) and *VACHT* (vesicular acetylcholine transporter) are generally co-expressed in cholinergic neurons. Although *VACHT>CFP* also labeled both distal neural rings, the zone of expression appeared to be limited in comparison with *PC2>Kaede* (S5C and S5D Fig). *VACHT>CFP* positive cells were localized within a proximal narrow band exhibiting high levels of reporter expression along with a more distal dispersed band exhibiting lower levels of reporter expression. We next used a *TH>Kaede* reporter to label potential dopaminergic or noradrenergic neural-like cells. Strikingly, this reporter was only expressed in the distal portion of the ventral cardiac plexus (S5D and S5E Fig). Using double *TH>-Kaede*, *VACHT>CFP* lines, we were able to discern that Kaede (TH+) and CFP (VACHT+) cells do not overlap within the ventral plexus (S5D Fig). Thus, these distal TH+ cells may correspond to the NP-C sub-cluster which displays enriched expression of *TH* but not *VACHT* (Fig 3I).

## Developmental profile of cardiac intrinsic innervation in *C. robusta* juvenile hearts

We next employed *PC2>Kaede*, *VACHT>CFP,* and *TH>Kaede* stable transgenic lines to analyze initial innervation of juvenile hearts (S6 Fig). In early juvenile samples (D3) a single *VACHT>CFP*+ cell was observed at both the ventral and dorsal ends of the heart (S6A Fig). In double transgenic *VACHT>CFP–PC2>Kaede* early juveniles (D4), we were able to observe multiple distinct VACHT−/ PC2+ or VACHT+/ PC2− or VACHT+/ PC2+ cells (S6B Fig). By day five, the *VACHT>CFP*+ ventral plexus appeared to consist of two distinct, large foci (S6C Fig). As seen in the adult heart, *TH>-Kaede*+ cells are restricted to the ventral plexus in early juvenile hearts and do not colocalize with *VACHT>CFP*+ cells in double transgenic *VACHT>CFP*, *TH>Kaede* juveniles (S6D Fig). In D7 juveniles, we were able to observe extensions from the ventral *TH>Kaede*+ neurons that appear to innervate adjacent progenitors in the distal UL (S6E and S6F Fig). In older juveniles (D10–D12), *PC2>Kaede*+ neurons form a small plexus at each end of the heart (S6G and S6H Fig). As seen in earlier stages, these plexuses are in close proximity to the distal UL clusters and exhibit thin neurite-like projections

associated with the adjacent line of cardiomyocyte progenitors (S6H Fig). To more closely examine the relative positions of these two cell types in the adult heart, we dissected and flattened distal growth zones in transgenic *PC2>Kaede* adults. In these samples, PC2+ neurites appear to be interspersed within the densely clustered nuclei of the distal growth zone (S6I and S6J Fig). In summary, our reporter analysis combined with our scRNA-seq data indicates that there is a complex array of neural-like cell types in these early distal plexuses and suggests that there may be shifts in the composition of these cardiac neural-like cells as the heart matures. Our reporter data also reveals that neural-like cells are associated with the distal ends of the forming myocardial tube in early juveniles, just as the heart is completing morphogenesis. Intriguingly, some of these early juvenile neural-like cells appear to innervate distal cardiomyocyte progenitor clusters located on either end of the UL.

## Assessing the role of neural inputs in regulation of *C. robusta* heart rate

The presence of VACHT+ and TH+ cardiac neural-like cells indicates that heart rate could be modulated by cholinergic or dopaminergic/noradrenergic inputs, respectively. Additionally, highly enriched expression of the sole *C. robusta* ortholog to the *tachykinin receptor* (*TACR*) in the presumed cardiac neural-like cluster (Cluster 8, Fig 3C) indicates that *C. robusta* heart function could be modulated by tachykinin signaling. To test these hypotheses, we quantified heart rate in D5–D6 juveniles both before and after exposure to a set of drugs targeting these pathways (S7A Fig). Application of the TH inhibitor α-methyl-para-tyrosine (AMPT) lowered heart rate slightly, but this result was not significant. Furthermore, application of the acetylcholine agonist acetylcholine chloride and the TACR inhibitor Aprepitant had no discernible impact on heart rate. Based on these results, we began to investigate other potential roles for these neural signals.

## Assessing the role of tachykinin signaling in heart growth

Based on the proximity of PC2+ neural-like cells to the distal progenitors at each end of the UL in juvenile and adult hearts (S6 Fig), we hypothesized that these neural-like cells may regulate distal progenitor cell behaviors. Previous studies have shown that tachykinin family members substance P and substance K promote cell proliferation in cultured vertebrate muscle cells [142]. Additionally, a more recent study found that substance P promotes cell proliferation in cultured cardiomyocyte progenitors [141]. Thus, we hypothesized that tachykinin signaling may play a similar role in *C. robusta*, promoting heart growth by increasing proliferation of cardiomyocyte progenitors in the distal UL growth zones.

To begin investigating a potential role for either tachykinin or cholinergic signaling in heart growth, we injected *C. robusta* adult hearts with the TACR inhibitor Aprepitant, the cholinergic inhibitor Atropine, or with control carrier solutions and cultured the otherwise intact animals for 24 hours. In order to assay myocardial proliferation, hearts were removed after drug treatment and incubated in EdU. As predicted by our hypothesis, hearts treated with the TACR inhibitor displayed a significantly lower percentage of EdU+ cells in the distal region in comparison with matched controls (Fig 5A–5D), whereas inhibiting cholinergic signaling by injecting Atropine did not impact cell proliferation (Fig 5D). However, it was difficult to discern which cell populations were impacted. To more precisely examine the role of tachykinin signaling on heart growth, we treated D3 juveniles for four days with one of two distinct tachykinin receptor antagonists, Aprepitant or Spantide II, and assayed the number of midline progenitors in the resulting D7 juveniles. As predicted by our hypothesis, both Aprepitant and Spantide II significantly reduced the number of midline progenitor cells in comparison to controls (Fig 5E–5I). To explore the specificity of this result, we treated juveniles with inhibitors against a number of other neural signaling pathways including the acetylcholine receptor antagonist Atropine, the alpha-1 adrenergic receptor antagonist Doxazosin Mesylate, and the TH pathway inhibitor AMPT. None of these drugs had any discernible impact on heart growth or midline progenitor numbers (S7B–S7H Fig). We also performed a gain-of-function experiment, treating D3 juveniles with Ramipril, an inhibitor of the angiotensin-converting enzyme pathway that breaks down tachykinin [143]. As predicted by our hypothesis, treatment with Ramipril resulted in a robust and significant increase in the number of midline progenitors in comparison to controls (Fig 5J–5L).

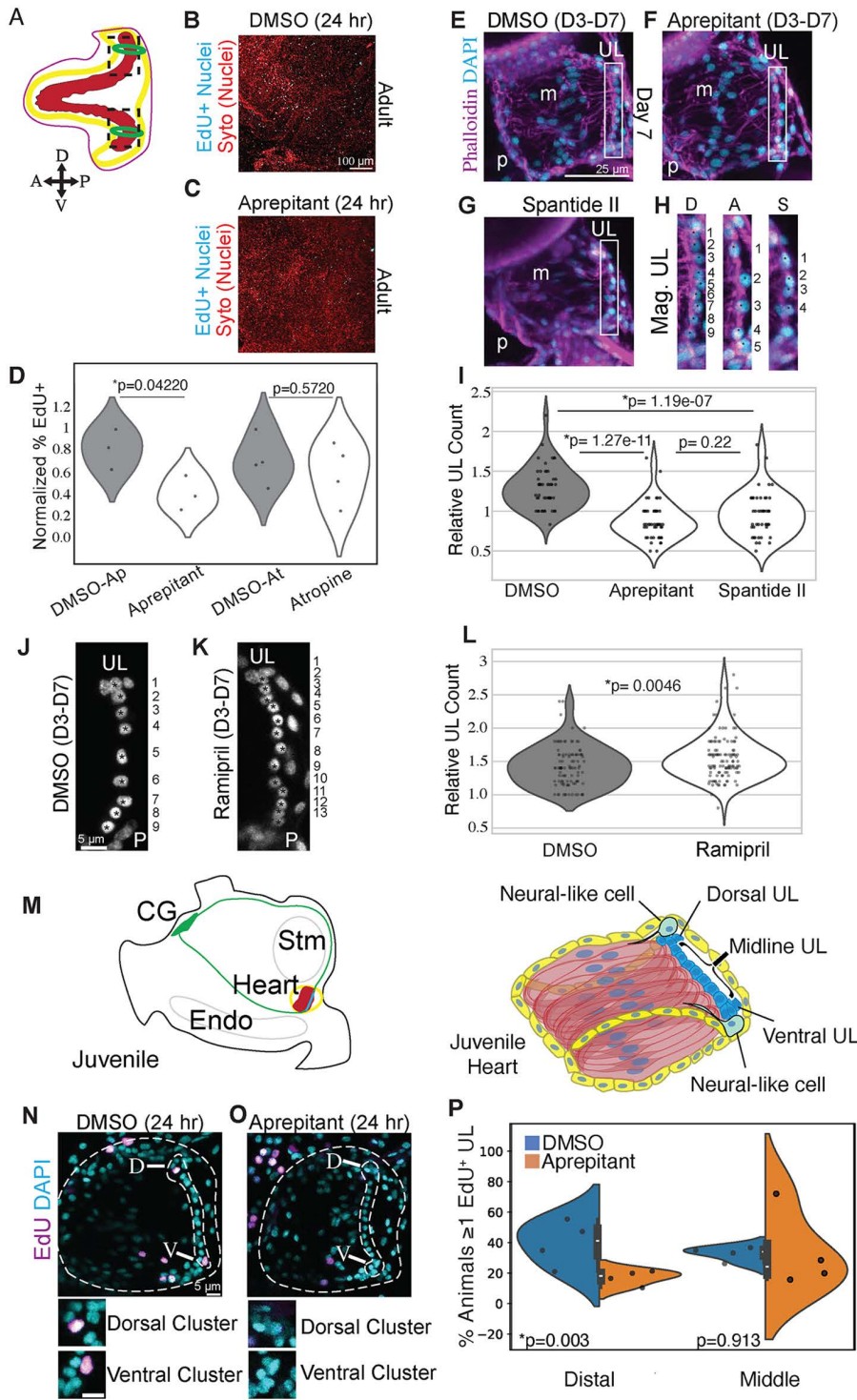

**Fig 5. Tachykinin signaling promotes distal progenitor proliferation. (A)** Diagram of adult heart. Boxes indicate distal areas shown in panels **B** and **C**, green rings indicate distal neural plexuses. **(B, C)** EdU (cyan) labeled distal UL section of control adult heart or an adult heart treated with Aprepitant for 24 hours as labeled. **(D)** Violin plot quantification of percent EdU+ cells per treatment; $n = 3$ Aprepitant hearts, $n = 4$ Atropine hearts. **(E)** D7 control heart. **(F)** D7 heart treated with Aprepitant. **(G)** D7 heart treated with Spantide II. **(H)** Enlargements of UL cells from boxed areas of **E**, **F**, and **G**. In **E–H**, asterisks mark individual UL nuclei in magnified panel. **(I)** Violin plot quantification of normalized midline UL cells per treatment. Midline UL count normalized to control per trial. Three trials per condition. $N = 49$ DMSO, $N = 49$ Aprepitant, $N = 46$ Spantide II. **(J, K)** Representative UL regions from hearts in D7

juveniles that were treated with DMSO or Ramipril for 4 days as labeled. **(L)** Violin plot of UL count distributions in control vs. Ramipril-treated samples. Midline UL count normalized to control per trial. 4 trials per condition, $N = 99$ DMSO, and $N = 114$ Ramipril. **(M)** Diagram of relative cardiac and neural anatomy in a juvenile (left) and diagram of juvenile heart anatomy (right). **(N)** D11 control DMSO-treated heart. **(O)** D11 heart treated with Aprepitant for 24 hours. In **N** and **O**, outline of the heart and UL depicted by dashed lines. Cyan indicates DAPI staining and magenta indicates EdU-labeled nuclei. **(P)** Split-violin plot of percent animals per trial with one or more EdU+ progenitor in the distal or midline UL per trial. Four trials each condition, $N = 228$ DMSO, $N = 213$ Aprepitant total animals across trials. For **D**, a $t$ test was performed between the averages per trial. For **I**, **L**, and **P**, a Shapiro–Wilk test determined that data was not normally distributed, thus a Mann–Whitney $U$ test followed by Bonferroni correction was performed. The data underlying the graphs in this Figure can be found in S1 Data.

To more directly assay the impact of tachykinin pathway inhibition on cardiomyocyte progenitor proliferation, we treated D10 juveniles with Aprepitant or DMSO for 24 hours and labeled proliferating cells using a 6-hour EdU pulse (Fig 5N–5P). As predicted by our hypothesis, we observed a significant reduction in the number of distal EdU+ progenitors in comparison to controls (Fig 5N–5P). In contrast, although the number of EDU+ midline progenitors were more variable in treated samples, no reduction was observed in this population. Taken together, our data strongly supports a role for tachykinin signaling in promoting distal cardiomyocyte progenitor proliferation.

## Expression patterns for the genes encoding tachykinin and its receptor

Our analysis of the scRNA-seq data indicates *TACR* is uniquely expressed in the neural-like sub-cluster (Cluster 8, Fig 3C). Based on this data, we hypothesized that tachykinin signaling impacts heart growth indirectly, impacting VACHT+ neural-like cells which then directly regulate proliferation in nearby distal progenitors (Fig 6A). To begin investigating this hypothesis, we used fluorescent in situ hybridization (FISH) to visualize *TACR* expression in juvenile hearts. Although the TACR expression pattern was often obscure, we did occasionally observe expression in cells adjacent to distal progenitors (S8A Fig). We next began to explore the expression of the sole *C. robusta* ortholog to the vertebrate tachykinin family neuropeptides (*TK*). Previous studies have documented that *TK* is expressed exclusively in the brain, intestine, endostyle, and gonads [144]. In line with this previous data, we did not detect expression of *TK* in any of the cardiac or blood cell clusters in our scRNA-seq dataset (S8B Fig). We therefore employed a Tachykinin reporter (*TK>CFP*) to document the source of tachykinin in D7 juveniles. As predicted by our scRNA-seq data, the TK reporter was not expressed in the heart (S8C and S8D Fig). Instead, in line with previous studies, we observed reporter expression in the central ganglion (S8C Fig).

## Characterization of central ganglion neurons innervating the *C. robusta* heart

According to our current model, tachykinin produced by central ganglia cells interacts with TACR expressed by cardiac neural-like cells associated with the distal pools of cardiac progenitors (Fig 6A). In vertebrates, cholinergic extensions of the vagal nerve innervate the heart [72,145]. Previous examination of the *C. robusta* nervous system did not reveal any innervation from the central ganglion to the heart [98,146]. To explore potential extrinsic innervation of the *C. robusta* heart by the CNS, we used high-intensity confocal laser settings to visualize neurons labeled by *PC2>Kaede* in the hearts of young *C. robusta* adults and D10 juveniles (Fig 6). This analysis revealed a network of thin *PC2>Kaede*+ projections that appear to branch off the relatively thick projections of the central ganglia and extend over a large surface area of the heart in both adult and juvenile stages. Although these presumably extrinsic neural projections were highly variable in morphology they consistently appeared to connect with dorsal and ventral *PC2>Kaede*+ neural plexuses in both young adults and D10 juveniles (Fig 6B–6D). The ventral (hypobranchial) cardiac neural plexus appears to be innervated by neurites extending from central ganglion axons located along the longitudinal muscle band on the right side of the animal. The dorsal (visceral) cardiac neural plexus appears to be innervated by neurites extending from central ganglion axons that wrap around the posterior end of the stomach before reaching the heart.

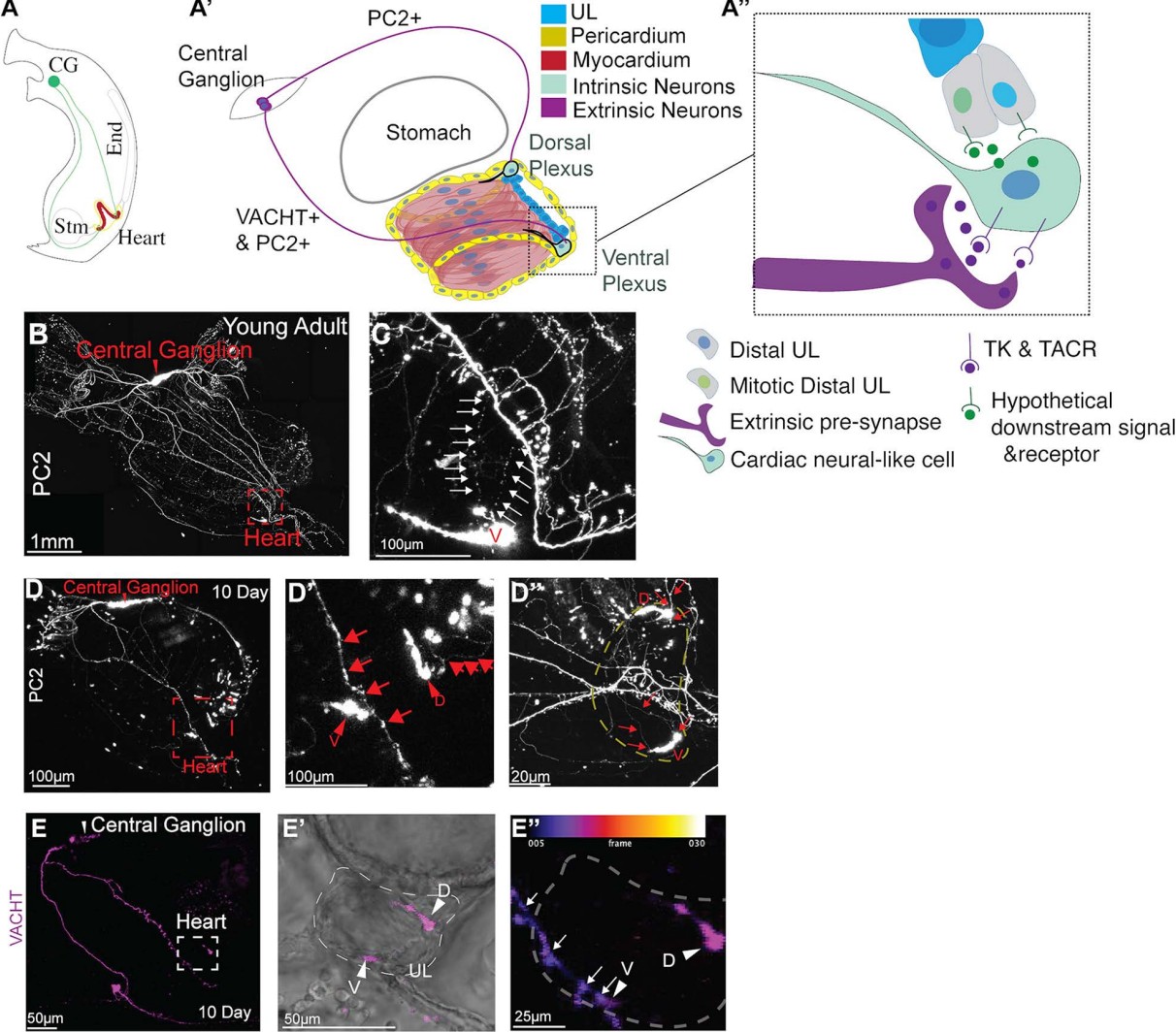

**Fig 6. Neurites extending from the central ganglia appear to innervate cardiac distal neural plexuses. (A)** Diagram of cardiac and neural anatomy in an adult. Green lines denote possible extensions of the central ganglion to each side of the heart. (**A′**) Diagram of cardiac innervation in a juvenile heart. Extrinsic neurons (purple) from the central ganglion connect to dorsal and ventral cardiac neural plexuses (green). (**A″**) Schematic model in which tachykinin secreted by central ganglion neurites is received by TACR-expressing cardiac neural-like cells (either intrinsic neurons or pacemakers), causing them to secrete a hypothetical downstream signal that promotes proliferation of adjacent distal myocardial progenitors. (**B, C**) Young adult displaying *PC2>Kaede* expression (white). **C** shows the magnified boxed area from **B**. Arrows highlight the connection between central ganglion projection to the ventral cardiac neural plexus. (**D**) *PC2>Kaede* expression in a D10 juvenile (white). **D′** shows the magnified boxed area from **D**. Arrows indicate connection from central ganglion projection to ventral and dorsal cardiac neural plexuses (arrowheads). **D″** shows a heart from a D30 juvenile displaying the same connections as in **C** and **D**. Yellow dashed line outlines the heart. (**E**) D10 transgenic juvenile displaying *VACHT>CFP* reporter expression (magenta). **E′** shows the magnified boxed area from **E**, including brightfield to reveal the position of the heart. **E″** Depth-encoded view of the same heart reveals close proximity between a central ganglion neurite (arrows) and a cell in the VACHT+ ventral cardiac neural plexus (arrowhead).

To further explore potential similarities between vertebrate and *C. robusta* cardiac innervation, we determined whether extrinsic neurons that appear to innervate the distal intrinsic plexuses of the *C. robusta* heart are cholinergic using the stable *VACHT>CFP* reporter lines. As observed previously, the *VACHT* reporter labeled cardiac neural plexuses on both ends of the heart (Fig 6E and 6E′). Additionally, we observed a network of extrinsic cholinergic neurons (*VACHT>CFP+*) that extended along the longitudinal muscle and appeared to innervate the ventral intrinsic plexus (Fig 6E, 6E′ and 6E″).

## Targeted knockdown of either *tachykinin* or the *tachykinin receptor* severely disrupts formation of the myocardium

To more definitively disrupt cardiac tachykinin signaling we used CRISPR-Cas9 to knockdown either *tachykinin (TK)* or the *tachykinin receptor* (*TACR*) and examined the impact on juvenile heart growth. Based on our model we predicted that both manipulations would reduce distal progenitor proliferation, leading to decreased heart growth. To knockdown *TK*, we used the *PC2* enhancer to drive Cas9 expression in the PC2-positive neurons [98] and co-expressed a pair of *TK*-specific sgRNAs ubiquitously using the *U6* promoter [147]. As documented above, the PC2 enhancer drives reporter expression in both extrinsic central ganglia neurons and distal cardiac plexuses (Fig 6); however, *TK* is only expressed in the central ganglia (S8 Fig). Thus, we used this approach to determine whether *TK* produced by extrinsic neurons is required for heart growth. As predicted by our model, we found that a significantly higher proportion of D10 *TK* crispant juveniles displayed abnormal heart morphology in comparison to controls (Fig 7A and 7B, and S1–S2 Movies). The partial penetrance of this phenotype aligns well with other CRISPR-Cas9 assays in *C. robusta* and is considered to reflect mosaic incorporation of plasmids [148–151]. Additionally, partial penetrance may result from naturally occurring sequence variation in the targeted region (S8 Fig) [152]. Surprisingly, abnormal hearts were not markedly reduced in size but instead appeared to consist of a nearly hollow pericardial cavity with no discernible UL and in which beating myocardial cells were either completely absent or restricted to a small rudiment (Fig 7A and 7B and S1–S2 Movies). High-resolution confocal imaging confirmed this phenotype (Fig 7C and 7D). To knockdown *TACR*, we used the *Mesp* enhancer to drive *Cas9* expression in cardiac lineage cells and co-expressed a *TACR*-specific sgRNA ubiquitously using the *U6* promoter. As observed in the *TK* knockdown assay, loss of *TACR* severely disrupted heart morphology, generating a similar hollow pericardial cavity phenotype (Fig 7B, 7E, and 7F and S3–S4 Movies). As mentioned in relation to *TK* knockdown, partial penetrance of this phenotype likely reflects mosaic incorporation of plasmids [149]. Additionally, partial penetrance may result from population-specific sequence variation in the targeted region (S8 Fig, [152]). Strikingly, loss of the myocardium in these *TACR* mutant hearts was associated with the absence of PC2+ distal neural plexuses (Fig 7G and 7H). Due to the severe nature of the hollow pericardial phenotype, it was difficult to assess the presence of the UL. To determine whether any remnant of the UL was still present in these abnormal hearts, we visualized the expression of the UL marker gene *CrUL1* in TK mutant hearts. While all control hearts displayed strong *CrUL1* expression just posterior to the UL (Fig 7I), all abnormal TK mutant hearts lacked any discernible *CrUL1* expression (Fig 7J). These data provide strong support for our model, indicating that both tachykinin secreted by central ganglia neurites, and tachykinin receptors in cardiac neural-like cells, are required for cardiac progenitor proliferation. Indeed, these assays indicate that tachykinin-dependent progenitor proliferation is not only required for heart growth, but also plays a critical, earlier role in formation or maintenance of the myocardium.

## Discussion

In this paper, we have begun to establish the *C. robusta* heart as a platform for investigating the contributions of paracrine and neural signals to cardiogenesis along with subsequent growth. In particular, our data indicate that both canonical Wnt and tachykinin signaling play critical roles in heart formation and growth (Figs 4 and 7). We first provide lineage tracing data indicating that the UL serves as a stem-like progenitor population capable of producing new myocardial cells (Fig 2). Through clonal analysis, we delineated distinct division patterns in distal versus midline progenitor population and generated a model regarding the role of asymmetric divisions in sustaining a distal reserve stem population while promulgating elongation of the UL and myocardial tube (Figs 2 and S2). Through scRNA-seq of the adult heart we identified eight distinct cell states, including a cluster of cells that represent the UL (Figs 3 and S4). Analysis of expression patterns within this presumptive UL cluster led us to investigate the role of canonical Wnt signaling, revealing that cWnt negatively regulates proliferation in the UL, potentially contributing to the low proliferative index of these cells and constraining heart growth (Figs 3 and 4). We also began to investigate a cardiac neural-like cell state that may include both cardiac intrinsic

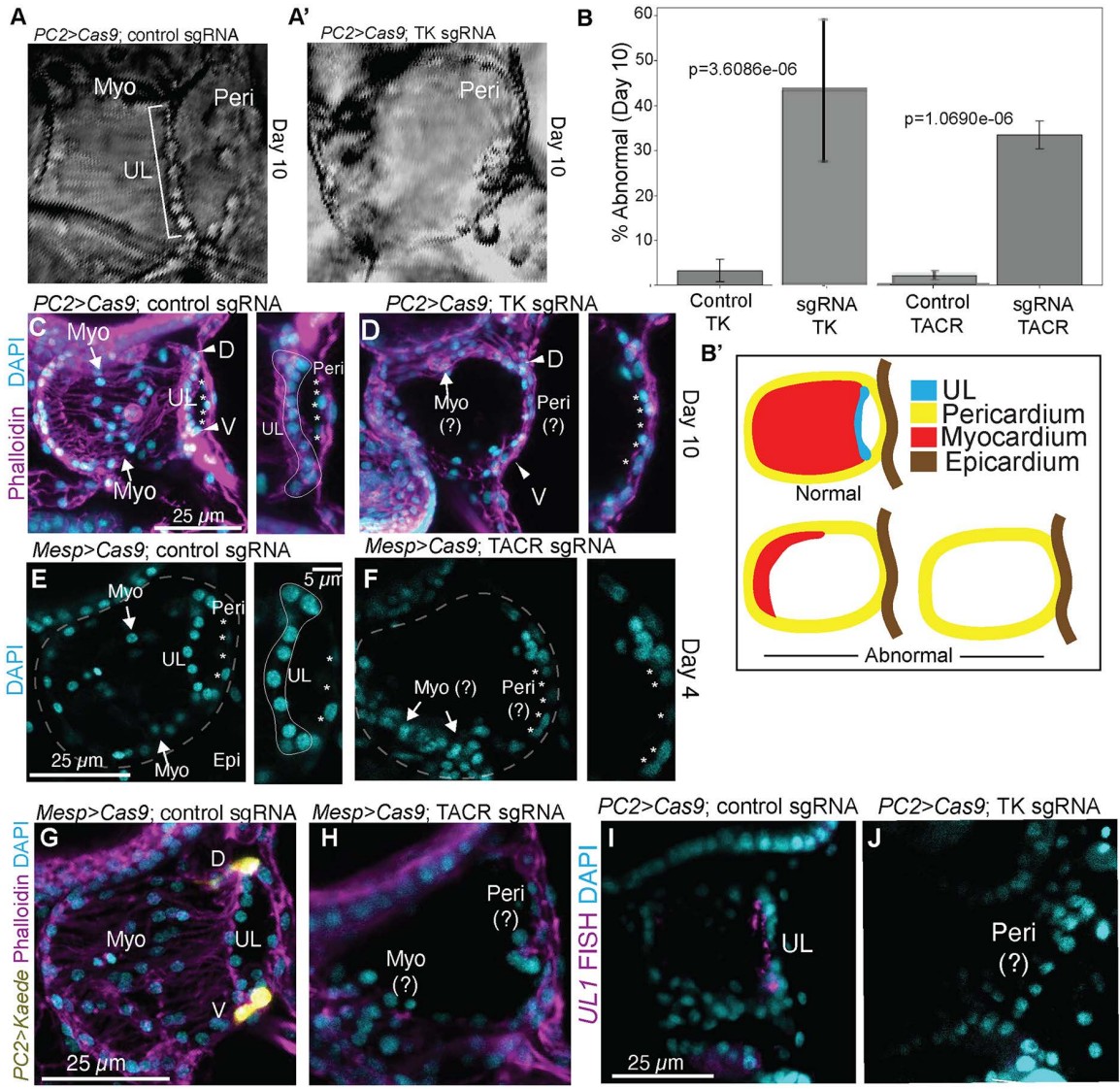

**Fig 7. Targeted knockdown of tachykinin in extrinsic neurons or of tachykinin receptor in cardiac lineage cells abrogates formation of the myocardial tube. (A)** Representative transmitted light images of a D10 control heart (**A**) and a CRISPR-Cas9 TACR crispant heart (**A′**). Note the hollow pericardium (labeled as "Peri" in this and subsequent panels) lacking a discernible myocardial tube (labeled as "Myo" in this and subsequent panels) or UL in the crispant heart. **(B)** Bar graphs of treated and control hearts scored for morphology. When targeting *TK*, 2/64 control hearts were abnormal and 24/64 TK crispant hearts were abnormal. When targeting *TACR*, 1/60 control hearts were abnormal and 22/70 TACR crispant hearts were abnormal (three trials per group, p-values displayed in graph). **(B′)** Schematic illustrating normal and abnormal heart phenotypes. **(C)** Representative micrograph of D10 juvenile control heart showing normal heart anatomy including myocardial tube, pericardium, and UL. **(D)** Representative micrograph of a *TK* crispant heart showing lack of normal myocardial tube and UL. In this and subsequent panels, "Myo (?)" Indicates the presence of a cluster of cells that may be myocardial while "Peri (?)" indicates presumptive pericardium. **(E)** Representative micrograph of a D4 control juvenile heart stained with DAPI. Panel on right is magnified showing UL (dotted line) and pericardial nuclei (asterisks). **(F)** Representative micrograph of a TACR D4 crispant heart showing lack of normal myocardial and UL cells within the pericardium (asterisks). 8/8 TACR crispants with abnormal heart morphology had no UL cells, 3/3 controls had normal UL morphology. **(G)** D10 control juvenile electroporated with *PC2>Kaede* (yellow) labeling the dorsal and ventral intrinsic neuronal-like clusters (D, V). **(H)** Representative D10 TACR crispant demonstrating lack of *PC2>Kaede+* cells in the heart in otherwise PC2+ juveniles. 8/8 TACR crispants with abnormal morphology had no PC2 reporter in the heart, 3/3 controls had normal PC2 expression. **(I)** Representative control showing *CrUL1* expression (magenta) associated with the UL in a D7 juvenile heart, all 21 samples displayed normal *CrUL1* expression across two trials. **(J)** Representative D7 TACR crispant showing lack of *CrUL1* expression (magenta) as observed in 11/26 samples across two trials, as mentioned above these 11 samples had abnormal heart morphology while the other 15 had normal hearts. The data underlying the graphs in this Figure can be found in S1 Data.

neurons and pacemaker cells (Figs 3 and S5). Using high-resolution microscopy and reporter constructs, we mapped distinct cardiac neural sub-types in both adult and juvenile *C. robusta* hearts (Figs 6 and S5). Remarkably, we found that distal neural plexuses appear to innervate adjacent pools of distal progenitors (Figs 6 and S6). Based on high levels of *tachykinin receptor* (*TACR*) expression in this cluster, we investigated the role of this neuropeptide. Our findings indicate that tachykinin signaling promotes proliferation of distal myocardial progenitors (Figs 5 and 7). According to our current model (Fig 6A″), tachykinin secreted by central ganglion neurites stimulates cardiac intrinsic neural plexuses, which in turn produce an uncharacterized signal required for heart morphogenesis and growth.

## Progenitor and precursor lineages contribute to growth of the *C. robusta* myocardium

Our data suggests that new myocardial cells are derived from three distinct populations - distal and midline progenitors that reside in the UL along with myocardial precursors that migrate anteriorly before differentiating. Furthermore, we posit that both distal and midline progenitors behave like cardiac stem cells, dividing asymmetrically to self-renew while producing a midline or precursor daughter respectively (Fig 2H). We recognize that our data in support of these presumptive asymmetric divisions is limited. Future work will focus on characterizing the distinct transcriptional profiles and developmental origins of these presumed progenitors. The resulting data will allow us to more rigorously track the division patterns for each of these UL sub-types. Additionally, these data will be used to investigate whether either UL sub-type exhibits patterns of gene expression or cell behavior reminiscent of characterized stem cells in other organisms, or if they bear any notable similarities to cardiac progenitor populations that reside at the distal ends of the vertebrate embryonic heart tube [153–155]. We are also interested in discerning whether there are niche-like structures that maintain stemness in these populations. Along these lines, it is intriguing that the anterior side of the UL exhibits enriched actin, WGA, and localized accumulation of *CrUL1* mRNA (Figs 1 and S4). These features lead us to hypothesize that interactions of the UL with myocytes or the ECM on this anterior side may serve to promote stemness. In line with this hypothesis, the division of midline progenitors perpendicular to the UL axis may allow only one of the daughters to remain in contact with this presumed anterior niche. It is important to point out that our model of heart growth is primarily based on data from juvenile hearts. Widespread mitosis within the adult myocardium (Fig 1L and 1M) suggests that differentiated myocardial cells may be capable of undergoing mitosis or dedifferentiating into precursors. Alternatively, a population of precursors may remain embedded in the myocardium. Although we did not observe any cells in the myocardium that exhibited distinctive morphological features indicative of a precursor, further studies are needed to address this point.

## The role of canonical Wnt signaling in heart growth

Though our data indicate that canonical Wnt signaling is required to suppress UL proliferation (Fig 4), the underlying mechanisms remain to be explored. Further studies will focus on determining whether cWnt signaling acts directly to drive progenitors into mitotic arrest or if it acts indirectly, for instance by promoting myocardial differentiation of progenitor daughter cells or by fate specification into another less proliferative cell type. In vertebrate embryos, Wnt-dependent contributions to heart development are complex and vary depending on the model organism, impacted cell population, and developmental stage [156–158]. A more coherent understanding regarding the role of cWnt signaling in *C. robusta* heart growth will allow productive comparisons with vertebrate cardiogenesis, potentially delineating specific functional similarities. One of the most exciting results of this part of our study was that inhibition of cWnt signaling led to an overall increase in heart size without apparently impacting heart morphology. Thus, we anticipate that insights into the role of cWnt signals in *C. robusta* heart growth will inform a broader understanding of proportional organ growth.

## A complex range of neural and neural-like cell types are associated with the *C. robusta* heart

Our data suggest that the *C. robusta* heart is innervated by extrinsic neurons originating from the central ganglia that connect with two previously described distal intrinsic neural plexuses located at either end of the heart (Fig 6). Our data

also suggests that these plexuses consist of diverse set of neural-like cells that may include both intrinsic cardiac neurons and pacemaker cells (Figs 3 and S5). Future work will focus on more thoroughly characterizing the cellular composition of these cardiac neural-like cell populations and on performing rigorous functional analysis to discern their contributions to cardiac function or growth.

### Neural control of progenitor cells proliferation and organ growth

The most striking finding in this study relates to the presumed role of cardiac neural-like cells in promoting cardiomyocyte progenitor proliferation (Figs 5 and 7). Although our studies strongly support a substantive role for tachykinin signaling in neural-dependent heart growth, many key aspects of our model remain ambiguous. One major gap relates to the specific identity of cells that respond to tachykinin signaling. To overcome this gap, it will be critical to perform a more robust characterization of *TACR* expression in adult and juvenile hearts along with targeted CRISPR-Cas9 knockdown of *TACR* in distinct populations of cardiac neural-like cell types in order to identify which cells mediate tachykinin-dependent cardiac progenitor proliferation. Another major gap relates to the identity of the proliferative signal produced by cardiac neural-like cells. Based on signaling factor expression in the scRNA-seq cardiac progenitor cluster (Fig 3F), we are particularly interested in investigating potential downstream roles for the RA, EGF, Insulin, Notch, and TGF-beta pathways. We are also interested in investigating whether TK signaling instructs neural-like cells to secrete cWnt inhibitors, thereby blocking Wnt-dependent suppression of progenitor proliferation (Fig 4) in order to promote heart growth. Lastly, a role for neural signaling in cardiomyocyte progenitor proliferation suggests that environmental or physiological conditions could mediate changes in heart growth by modulating extrinsic neural inputs. This hypothesis could be tested by examining the impact of nutritional availability or other external factors on heart growth and determining whether any such impacts require extrinsic cardiac innervation.

Though many studies have revealed roles for innervation in promoting organ development, few studies focus on the impact of innervation on progenitor or stem cell proliferation [12,13,16]. Literature on potential roles for cardiac innervation or neuropeptides in modulating cardiomyocyte progenitor proliferation generally focus on post-natal growth or injury [23,81,141,159]. Thus, insights gained from further exploration of heart formation and growth in *C. robusta* may inform efforts to better understand the role of tachykinin and other neural signals in vertebrate heart development.

## Methods

### EdU

EdU staining was performed using the manufacturer's protocol (Invitrogen Click-iT). Briefly, juveniles or adult heart tissue was incubated in filtered artificial seawater (FSW) and the EdU staining solution in 35 mm dishes and reared at 18 °C for the desired amount of time (between 30 min and 24 hours). Samples were rinsed 3 times with (FSW) and then either pulled for fixation or left for a pulse phase. EdU-incorporated samples were fixed in 4% PFA in FSW overnight at 4 °C, rinsed twice with PBS, and incubated in the Click-iT reaction buffer for 30 min. Samples were then rinsed twice with PBS, incubated in 1× DAPI/PBS for 20 min at room temperature, then rinsed in PBS prior to mounting.

### WGA, Phalloidin, and DAPI staining

Juveniles were first relaxed by adding a small (~3 mm length) crystal of menthol directly to a 60 mm dish of FSW for 15–30 min and were monitored until unresponsive. Juveniles were then fixed using 4% PFA in seawater for 15 min at room temperature. Juveniles were then washed twice with PBS. To permeabilize, juveniles were incubated for 1 hour in 0.5% Triton-X in PBS while rotating on a nutator. Following permeabilization, juveniles were incubated in DAPI (Thermo Fisher 62248) (1:1000) and Phalloidin (1:100) overnight at 4 °C on a nutator covered in foil. Juveniles were then incubated for 5 min twice with PBS. Juveniles were mounted on slides in PBS using coverslip spacers. Samples receiving WGA staining were incubated in 10 µg/mL WGA-488 (Thermo Fisher W11261) for 1 hour on a nutator then rinsed twice with 1× PBS prior to the permeabilization step.

## Electroporation protocol

Gravid *Ciona robusta* were obtained from MREP and Marinus Scientific, collected from various locations in southern California. Animals were maintained under constant illumination in a recirculating refrigerated tank to promote gamete accumulation. For electroporation, gametes from a minimum of three animals were combined. Transgene DNA constructs in water were pooled and mixed to a final concentration of 0.77 M D-mannitol. DNA concentrations were kept at or below 125 μg/mL and performed as in [149]. Electroporated embryos were subsequently cultured at 18 °C in FSW supplemented with 10 U/mL penicillin and 10 μg/mL streptomycin (FSW + pen/strep) in gelatin-coated dishes.

## Juvenile growth

Protocols for fertilization, dechorionation, and electroporation were carried out as described. At ~21 hours post fertilization (post-hatching stage), larvae were moved into scratched dishes and incubated at 18 °C. Two days after settlement, metamorphosed juveniles were moved into culturing aquaria. Approximately 2 L of seawater from our adult *C. robusta* holding tank was added to 2 L of FSW. A small 5 V USB pump was added for circulation. Scratched dishes containing the D3 juveniles were transferred to the 4L aquarium and kept at 18 °C. The scratch dishes are engineered to hang vertically within the aquarium using Styrofoam secured to one end and an Eppendorf weighted down with glass beads secured to the opposite end. Aquaria were covered with foil to prevent evaporation. Every 2 days, 2 L of water was replaced with FSW, and juveniles were fed 0.5 mL MicroVert (Kent Marine) and 0.5 mL Roti-Feast (Reef Nutrition) along with replenished pen/strep and ampicillin.

## Adult heart in vivo pharmacological inhibition

Glass needles were prepared for injection by pulling capillary tubes with a needle puller. Either 25 μL of Aprepitant (Sigma SML2215), Atropine (Cayman 12008), or molecular grade DMSO plus Fast Green dye was loaded into the needle. The final working concentration was 20 μM for Aprepitant and 50 μM for Atropine. Micro-dissection scissors were used to make a short and shallow incision in the tunic superior to the heart. The epicardial layer and longitudinal muscles that surround the heart and stomach were kept intact. Only if Fast Green dye was visible in the interior of the heart was an injection considered successful. Injected animals were then incubated overnight in a 400 mL container with FSW + Pen/strep at 18 °C. Hearts were then dissected out and were incubated with EdU for 24 hours before fixation. Hearts were stained with EdU Click-iT kit (Invitrogen C10637) according to manufacturer's protocol and co-stained for DAPI. EdU- and DAPI-stained nuclei were quantified using a custom pipeline in CellProfiler [160,161].

## Single-cell sequencing

Entire hearts were dissected in dishes in ice-cold FSW in 100 mm dishes. Micro-dissection scissors were used to cut hearts into ~2 mm$^2$ pieces of tissue. All tissue pieces were transferred to an Eppendorf tube using forceps to minimize transfer of excess FSW. 100 μL of Papain solution (Thermo Fisher 88285E) + 1 mg/ml collagenase (Gibco 17100-017) + 0.5 mg/mL cellulase (Thermo Fisher J64019) (kept on ice prior to use) was added to tissue and subsequently left at room temperature for a 30 min incubation with occasional mixing. Fifteen-minute trituration was performed using a P1000 pipette set to 1,000 μL, using slow deliberate pipetting at room temperature. A cell mesh filter was used to strain out clumps of cells and debris (Falcon 352235). Twenty percent Fetal Bovine Serum was added to ice cold magnesium and calcium-free FSW to inhibit the protease. Samples were then spun at 800$g$ (~2,800 RPM) for 3 min at 4 °C. Cells were washed with ice cold Mg/Ca$^{2+}$ free FSW twice and resuspended in 100 μL magnesium and calcium-free FSW. A hemocytometer and trypan blue staining was used to count cells and assess cell death percentage. Cells were diluted to 700 cells/μL using magnesium and calcium-free FSW. 7.1 μL of cells were then added to 36.1 μL of 0.7M D-mannitol for subsequent steps. Cells were kept on ice until 3p Capture and library preparation by the University of Pennsylvania

Next-Generation Sequencing Core. Two replicates were performed. 3p Library Prep was performed by the University of Pennsylvania Next-Generation Sequencing Core and sequenced first on a MiSeq, and then a NextSeq 2000 resulting in 195M reads for ~8,000 captured cells.

## Kaede photoconversion

Electroporation was carried out with standard protocol using 100 µg of *Mesp>Kaede* per electroporation. Embryos were kept in the dark using foil. Juveniles were grown according to the juvenile culturing protocol until D5. FSW + menthol was prepared by adding menthol crystals up into the 5mL line of a 50mL falcon tube and then filled to the 50mL line with FSW, incubated for 30min, and then placed on ice. Individual juveniles were placed in a small imaging dish with the FSW + menthol. A Zeiss LSM confocal was used for photoconversion and imaging using 40×. The stage was cooled to 4 °C. Zeiss Smart Setup was used to select Kaede-green and Kaede-red and transmitted light (PMT-T). A range of ~1%–10% laser for 488nm and 5%–10% 568nm laser as used with 650 V gain for the fluorescent PMTs, and ~150−230 V gain for the transmitted light PMT. Photobleaching using the ROI tool in Zen was used, with the 405nm laser for conversion and set to 5%. Cells were converted until the green signal was no longer visible (~10 to 30s). A 96-well plate was then used to culture individual juveniles (1 juvenile/well) after washing juveniles with fresh FSW+pen/strep (Gibco 15070-063). Animals were fed and imaged once every 24 hours for several days after initial photoconversion.

## Single-cell heart sequencing analysis

Reads were aligned to the *Ciona intestinalis* (*C. robusta)* Ciona3 genome assembly using gene annotations from the KH2012 gene models. Gene expression matrices were generated using the Cell Ranger count pipeline with default parameters and further processed using Scanpy [162]. Cells expressing fewer than 400 genes were filtered out. The data were then normalized to a target sum of 1e4 per cell and log-transformed using the log1p function. Technical effects were regressed out using regress_out, and data were subsequently scaled to unit variance with a maximum value of 10. Principal component analysis (PCA) was performed using the ARPACK solver. The log variance ratio was used to determine the number of informative principal components. A nearest neighbor graph was computed, followed by Uniform Manifold Approximation and Projection (UMAP) dimensionality reduction. Louvain clustering was performed on the neighborhood graph to identify transcriptional clusters. Two biological replicates were included and ingested together into the final displayed UMAP.

## Single-cell blood analysis

**Preprocessing.** A standard Scanpy pre-processing and clustering pipeline was used for each single-cell dataset. First quality control metrics were calculated to determine threshold values. An appropriate threshold was set to eliminate cells that did not meet the minimum number of genes. This threshold was dynamically set as determined by the depth and calculated distribution of the number of genes detected per cell; however, across all datasets a minimum of >200 genes/cell was required. Scatterplots of the number of genes by counts and total counts were generated to ensure dataset quality after the filtering step. Data were normalized to a target sum of 1e4 and then log-transformed to reduce variability of gene expression. $Log_e(1+x)$ was used, where $x$ is the normalized count. Regression was then applied to the data to reduce technical variability and batch effects across sequencing libraries. Finally, the data were scaled to a max value of 10 for intuitive comparison of gene expression.

**Dimensionality reduction.** PCA analysis was performed individually on the normalized datasets. The variance ratio was calculated and plotted to determine the ranking and impact each principal component contributed to the variance in the dataset, which was then used to determine the number of principal components to include for subsequent nearest neighbor analysis. UMAP was then used to visualize the sequencing datasets to visualize local and global structures in the datasets. Standard Louvain clustering was then performed on the UMAP. Manual inspection of known marker genes

was used as a quality control-step of Louvain and UMAP settings for stages where known markers are well established in the literature (see references therein).

**Elimination of blood clusters.** To identify blood cells, we compared our data to published scRNA-seq data of adult *C. robusta* circulating cells [101]. We identified cell clusters in the heart dataset which were transcriptomically similar to the blood dataset's published Leiden clusters. Specifically, for every pair of heart and blood dataset cluster pair, we calculated the correlation across highly variable genes of the mean log-transformed expression within each cluster. Cluster pairs with a correlation above a threshold of 0.5 are considered transcriptomically similar, and the heart clusters of those pairs were labeled as blood cells. This correlation threshold was chosen by running an equivalent analysis to compare biological replicates in the blood dataset, picking a threshold that identified cell states represented in both replicates.

Two manual adjustments were then made to the set of clusters which passed this threshold: (1) One cluster pairing, matching heart cluster 6 to blood cluster 10, has a correlation just under the 0.5 threshold (correlation = 0.494). Manual inspection of marker genes showed significant overlap, so we also labeled this heart cluster as blood cells. (2) One rare blood cell cluster, cluster 31 (0.17% of the observed blood cells) has a high correlation to heart clusters 1 and 8 (together, 12.1% of the observed heart cells). Given the large discrepancy in cell abundance, we did not label heart clusters 1 and 8 as blood cells. The final set of heart clusters matched to blood cell states is listed in S2 Table.

**Identification of marker genes and genes of interest.** Potential marker genes for adult *C. robusta* heart clusters were identified through a variety of methodologies to converge into a final list. First, known transcription factors were assessed for enrichment across clusters and then manually evaluated for biological relevance in each cluster. A Wilcoxon Rank-Sum Test was performed across clusters to identify genes of interest and potential marker genes in an unbiased manner, and then manually curated to highlight relevant genes. In addition, the top highest expressed genes were determined for each cluster and then manually selected depending on specificity. Last, specific and low-expressed genes were also identified for each cluster to identify any potentially important genes not identified through the other analyses. Known marker genes for cell types in *C. robusta* and across other species were used to begin to glean the cell type identity across the various heart clusters.

**Sub-clustering of adult *C. robusta* heart clusters.** After reporter and FISH analysis of various *C. robusta* stages (see section on cluster validation), the TACR cluster and UL clusters were further processed for analysis of sub-types of cells within each of these broader clusters. The TACR and UL clusters were subset in Scanpy, and then Louvain clustering was performed again on the UL and TACR cluster individually. The previously calculated PCA value for all adult *C. robusta* clusters was re-used to preserve the structure of the individual TACR and UL clusters. The Louvain clustering was recalculated for each new cluster using a resolution 0.83.

## In situ hybridization

In situ hybridizations were performed as in [128], with slight modifications for fluorescence in situ detection: After juveniles were phased out of hybridization buffer, they were washed three times with TNT (0.1 M Tris–HCl pH 7.4, 0.15 M NaCl, 0.1% Tween-20), then blocked for 1 hour in TNB buffer (0.1 M Tris–HCl pH 7.4, 0.15 M NaCl, 1% BSA). Juveniles were subsequently incubated overnight at 4 °C with a 1:1,000 dilution of POD-conjugated anti-DIG primary antibody (Roche). Following antibody incubation, embryos were washed three times with TNT and exposed for 5 min to FITC-tyramide working solution (Perkin Elmer) to fluorescently label antisense RNA probes. Juveniles were incubated in TNT for 10 min three times prior to mounting.

## Molecular cloning and reporter transgene information

*Syt15>GFP* was generated by PCR-amplifying approximately 1,500 bp of DNA upstream from the start of transcription using the forward oligo GGTGTTAAATTATCCACGATAAACCG and the reverse oligo TATCGTAGTATAACAACGA-CAAACTTGG. This fragment was subcloned into *Mesp>GFP* [97] after restriction digest removal of the *Mesp* enhancer fragment.

*Mesp>Kaede::NLS* (*Mesp>Kaede*) was generously provided by Alberto Stolfi. *Mesp>RFP* was previously described in [163].

*Tg[MiCiPC2K]2, Tg[MiCiPC2K]3, Tg[MiCiTnIG]2, Tg[MiCiTnIG]3, Tg[MiCib2TBC]3, Tg[MiCib2TBC]4, Tg[MiCiVACHTC]2*, were provided by Yasunori Sasakura (141,156–159), and *pCiTachykinint:CF* from Honoo Satake.

*VACHT>eCFP* cassette [164] of *pSPCiVACHTC* [165] was amplified by PCR using with PrimeSTAR HS DNA polymerase (Takara Bio Japan). The fragment was inserted into the BamHI site of *pMiLRneo* [166] by In-Fusion HD Cloning kit (Clontech) to create *pMiCiVACHTC*. *pMiCiVACHTC* was electroporated with Minos transposase mRNA to create stable transgenic line according to the previous method [167].

**FoxfΔepi>TCFdn.** Dominant-negative TCF (*TCFdn*), the *C. robusta* TCF coding sequence which lacks the N-terminus, was generated by PCR amplification of genomic DNA with the oligos CACCATGTACGATGTTCCGGCAAAAGTA and GCTGATGTTGCACGGCGG, as in [123]. *FoxfΔepi>TCFdn* was constructed by subcloning the *TCFdn* sequence into our *FoxfΔepi* vector (described in [128]) after restriction-digest removal of the *LacZ* sequence.

## Constructs for CRISPR-Cas9 targeting of *Tachykinin* and *Tachykinin Receptor*

To generate *PC2>Cas9*, we obtained *mMiCiPC2K* from CITRES (https://marinebio.nbrp.jp/ciona/top/top.jsp). *Mesp>-Cas9* was digested using AscI and NotI. The *mMiCiPC2K* enhancer was PCR amplified with added overhangs matching the *Mesp>Cas9* vector backbone using ACGTATTAATTAAGGCGCGTAACACCACGATATTAAATC and GGCTAGCCATGGTTGCGGCCCATTCAAATAAAATGCTGCT.

Listed sgRNA targeting sequences (see below) were obtained by using CRISPOR (https://crispor.gi.ucsc.edu/) [168] to maximize target efficiency and reduce off-target effects. Phosphorylated and annealed sgRNA oligos were cloned into an empty *U6>sgRNA* vector after digestion with BsaI as previously described in [169]. The sgRNA sequences used for targeting *Tachykinin* were TTGATGGGAAAACGATCAAT (*TKsgRNA2*) and CATCGTTCACTG-GCTTGATG (*TKsgRNA3*). The sgRNA sequences used for targeting *Tachykinin Receptor* (*TACR*) were AGTAACAA-GAGACGTGGGAC (*TACRsgRNA2*) and CGTATGGGCTCTGGGCGAAC (*TACRsgRNA3*). The control sgRNA used targeted GFP, as in [122]. Electroporation was carried out using 25 μg *PC2>Cas9* or *Mesp>Cas9*, 25 μg of each TK or TACR sgRNA pairs and 25 μg *PC2>Kaede* resulting in 100 μg total DNA per electroporation. *Mesp>Cas9* and the *U6>sgRNA* constructs (described in [170]) were generously provided by Lionel Christiaen. After experiments were conducted, the targeting efficiency of deployed sgRNAs were tested in larvae co-electroporated with *Ef1a>Cas9* along with either *TACRsgRNA1* or *TACRsgRNA2* and then subjected to Azenta Genewiz SNP/Indel analysis as described in [152]. Genomic DNA was harvested from a large pool of transgenic larvae (~500) and the region targeted by each sgRNA was then amplified by PCR and sent in for sequencing. For each sample, ~ 1,000–1,500 unique reads were analyzed for the presence of naturally occurring IN/DELs and for mutagenesis within the targeted locus. This analysis revealed a highly prevalent naturally occurring IN/DEL near both targeted loci. In these reads, there was a high level of sequence variation in the targeted locus. Thus, we manually filtered through the unique reads generated by this analysis for each sgRNA (*TACRsgRNA1* and *TACRsgRNA2*) to distinguish whether or not they contained the prevalent IN/DEL. We then assessed the incidence of mutagenesis within the unique reads that did not contain the IN/DEL. See the S8 Fig legend for additional details.

## Pharmacological inhibition of Tachykinin signaling

Immediately after metamorphosis, D3 juveniles were subjected to continuous pharmacological treatment until D7. In separate dishes, juveniles were either grown in 100 μM Aprepitant, 1 μM Spantide II (Sigma SCP0241), or control DMSO (Invitrogen C10337) for 4 days, followed by a 6 hr EdU pulse after the drug treatment. Juveniles were fixed and stained according to the above protocols and UL counts were assessed using FIJI [171].

## Pharmacological inhibition of neurotransmitters in juveniles

Juveniles were cultured according to protocol. Drugs were at the following concentrations: 100 µM AMPT, 9.13 µM Doxazosin Mesylate, or 50 µM for Atropine. Atropine, DMSO, or Doxazosin Mesylate was added to D3 juveniles for 4 days and UL length was counted in D7 animals after staining with DAPI and subsequent confocal imaging. AMPT or FSW control was added to D6 juveniles for 24 hours. Animals were fixed and stained with DAPI then imaged. UL counting was performed using FIJI [171].

## Pharmacological inhibition of the cWnt pathway

IWR-1-endo was dissolved in DMSO and applied at a final concentration of 2.5 µM. BIO (Sigma B1686) [126] was dissolved in DMSO and applied at a final concentration of 1.25 µM.

## Statistical analysis

For statistical analysis of TK and TACR crispant juveniles, a Fisher's Exact Test was performed using scipy.stats.fisher_exact. Analysis of normalized UL counts for Aprepitant, Spantide-II, Atropine, AMPT, and Doxazosin Mesylate was performed by first normalizing each trial by the counts for an animal in the DMSO condition and then taking the average normalized UL count of each trial. A *t* test was performed on the normalized trial values to obtain a p-value using excel. A *t* test was also performed using excel on the UL EdU anatomical region (ventral, middle, dorsal) measuring the average percent of animals per each trial that had one or more EdU positive cells per anatomical region of UL. For the adult heart growth by region, and for the injection assay of DMSO, Aprepitant, or Atropine, first the number of nuclei were counted in total using CellProfiler [161], then the number of EdU+ nuclei were counted to give a percent of cells that were EdU-positive. A *t* test in excel was performed comparing the percent of EdU+ cells per adult heart for each anatomical region (distal, middle, apical) or drug (DMSO, Aprepitant, Atropine). A paired *t* test was used on heart rate data before and after each drug treatment on individual juveniles for Aprepitant, Acetylcholine, and AMPT using SciPy.stats.ttest_rel in Python. To compare TCFdn and control hearts (Fig 4G), a permutation test was conducted to compare the proportions between treatment and control groups, showing significant difference under the level of 0.0001. (Fig 4G). Specifically, the *p*-value was computed based on permutation test comparing the average proportions of experiments with cell counts at least 1 SD greater than the corresponding trial's control mean.

## Supporting information

**S1 Fig. Juvenile midline UL and myocardial growth rates. (A)** Quantity of myocyte (blue) and midline UL cells (orange) from D3 to D15 using standard culturing conditions. **(B)** Quantity of myocyte (blue) and midline UL cells (orange) from D3 to D8 days using optimized culturing conditions (see Methods section). In **A** and **B**, D3-D8 X-axis labels are highlighted for comparison. Orange dots and lines, number of UL cells per time point.
(TIF)

**S2 Fig. Lineage tracing of myocardial progenitors in the UL using Kaede conversion.** In all panels, photoconverted cells (yellow) are tracked over multiple days in living juveniles starting with photoconversion at D5, timepoints indicate hours after conversion. See text for details. **(A)** Blue arrows track a single labeled midline UL cell that appears to divide symmetrically to produce two midline UL daughters. **(B, C)** Red arrows track single labeled midline UL cells that appears to divide asymmetrically to produce a midline UL daughter that remains in the same confocal plane as the rest of the UL, bottom row, along with a presumptive myocardial precursor that moves into a different confocal plane (top row). White arrows indicate neighboring cells in the UL. **(D)** Red arrows track a single labeled presumptive myocardial precursor that appears to migrate anteriorly, away from the UL. **(E)** Red arrows track a single labeled presumptive myocardial precursor that appears to mature over a 72-hour time-course as evidenced by elongation of the nucleus.
(TIF)

**S3 Fig. Pearson's correlation coefficient (PCC) analysis to assess cluster similarities.** PCC suggests Cluster 2 correlates strongly with Cluster 3 and that Cluster 1 correlates strongly with Cluster 7. Overlapping expression patterns between Clusters 1 and 7 may contribute to the lack of uniquely enriched genes in Cluster 1 (see main text and Fig 3A and 3B). Additionally, PCC revealed Clusters 5 and 8 did not correlate strongly with any other clusters suggesting these clusters are transcriptionally unique.
(TIF)

**S4 Fig. Characterization of presumptive epicardial and UL clusters by reporter or in situ expression analysis. (A)** Representative image displaying *Synaptotagmin 15* reporter (*Syt15>GFP)* expression in a D15 juvenile. *Mesp>RFP* labels pericardium, myocardium, and the UL (magenta). *Syt15>GFP* (green) was detected in the overlying epicardium along with a few large underlying cells which may be neural-like cells matching scRNA-seq data indicating that *Synaptotagmin 15* is expressed in the presumed epicardial and neural-like clusters (#4 and #8, Fig 3). **(B, C)** In situ expression patterns of marker genes associated with the presumptive UL cluster (#7, Fig 3). **(B)** *CrUL1* expression in a D5 heart. Colorimetric in situ hybridization (left), FISH (middle), and cartoon schematic (right). **(C)** *Gata.a* expression in a D8 heart. FISH (left) and cartoon (right). Note that in the micrograph a Z-plane containing myocardial cells in the anterior region of the heart is shown on the left while the Z-plane containing the UL in the posterior region of the heart is shown on the right. Blue **(B, C)** represents DAPI staining. Yellow **(B, C)** represents probe detection for each transcript. All images shown anterior to the left and dorsal up.
(TIF)

**S5 Fig. Location of cardiac neural-like/pacemaker cells in the adult *C. robusta* heart. (A)** Summary diagram of observed localization patterns for PC2+, VACHT+, and TH+ neural-like cells in the distal plexus of the adult heart. Note the ventral-exclusive presence of TH (green). **(B)** *PC2>Kaede* expression. **(C)** *VACHT>CFP* expression. **(D)** *VACHT>CFP* and *TH>Kaede* double-labeled adult hearts. Left column is *VACHT>CFP* (blue), middle column is *TH>Kaede* (green), and right column is merged. Middle and bottom rows correspond to enlarged areas of two different Z-planes of the boxed area in the top row. **(E)** *TH>Kaede* reporter expression at the distal end of the ventral plexus. Top panel is Kaede fluorescence, middle panel is brightfield, bottom panel is merged. Line in **E** indicates where the myocardial tube ends relative to the TH+ neural-like ring.
(TIF)

**S6 Fig. Location of cardiac neural-like/pacemaker cells in the juvenile heart. (A)** *VACHT>CFP* (red) in live D3 juvenile heart. **(B)** *PC2>Kaede* (red) and *VACHT>CFP* (blue) in a fixed D4 juvenile heart, outlined in white. **(C)** *VACHT>CFP* (red) in live D5 juvenile heart. **(D)** *VACHT>CFP* (blue) and *TH>Kaede* (yellow) in a fixed D5 juvenile heart, outlined in white. **(E)** *TH>Kaede* in a live D7 juvenile heart. **(F)** *TH>Kaede* (magenta) in a fixed juvenile heart. DAPI staining in cyan. **F′** shows a single Z-plane from **F**, note the overlap in magenta and cyan that appears to be associated with a UL cell (arrow). **(G)** DAPI stained nuclei of a D10 transgenic *PC2>Kaede* juvenile heart. **G′** shows *PC2>Kaede* expression in the same heart. **G″** shows the merged view. Note the overlap between *PC2> Kaede* expression and the distal UL clusters which is particularly clear at the dorsal end of the UL. **(H)** *PC2>Kaede* expression (red) in a D12 juvenile heart. Note staining at the dorsal and ventral ends of the UL as well as expression along the UL (within the dotted line). **(I, J)** *PC2>Kaede* labels cells that are interspersed with densely clustered DAPI-stained nuclei within the distal UL (outlined in white) in adult hearts. In both panels, the distal UL is distinguished by densely packed nuclei and is outlined by a dotted white line. *PC2>Kaede* is displayed in magenta (**I**) or white (**J**). **A**, **C**, **E**, and **H**, fluorescence merged with brightfield. In these images the UL is often outlined by a white dotted line and Myo indicates the position of the myocardial tube.
(TIF)

**S7 Fig. The impact of neural signaling modulators on heart rate and UL cell numbers. (A)** Violin plots of recorded heart rates in response to acetylcholine chloride (top), AMPT (middle), and Aprepitant (bottom). **(B)** D7 control heart. **(C)** D7 heart treated with Atropine. **(D)** D7 control heart. **(E)** D7 heart treated with Doxazosin Mesylate. **(F)** D7 control heart. **(G)** D7 heart treated with AMPT. **(H)** Violin plot of normalized UL counts per treatment. Counts normalized to the DMSO control for each trial. In **B–G**, cyan represents DAPI-stained nuclei. Images in **D–G** also show phalloidin staining (magenta). For **H**, a $t$ test was performed on normalized data, averaged across trials. Doxazosin Mesylate: $N = 60$ control and 58 experimental samples, 2 trials. Atropine: $N = 13$ control and 14 experimental samples, 2 trials. AMPT: $N = 46$ control, 33 experimental samples, 2 trials. The data underlying the graphs in this Figure can be found in S1 Data.
(TIF)

**S8 Fig. TK and TACR expression in the adult heart along with assessment of *TACR sgRNA* targeting efficiency. (A)** Fluorescent in situ hybridization of *TACR* expression. Blue represents DAPI and magenta displays fluorescent probe detection (left). Cartoon model of expression pattern (right), UL cells outlined in blue, myocardial cells outlined in red, green cells represent presumptive neural-like/pacemaker cells. **(B)** UMAP showing *TK* expression levels, note absence of expression. **(C)** Fluorescent *TK>CFP* reporter electroporation expression in the central ganglion (red). **(D)** Magnified region of the heart. Note absence of reporter expression, UL outlined in white, Myo indicates position of the myocardium. **(E)** Table displaying data generated from transgenic larvae co-electroporated with *Ef1a>Cas9* along with either *TACRsgRNA1* or *TACRsgRNA2* and then subjected to Genewiz SNP/Indel analysis. For each sample, ~1,000–1,500 unique reads were analyzed for the presence of naturally occurring IN/DELs and for mutagenesis within the targeted locus. This analysis revealed a highly prevalent naturally occurring IN/DEL near both targeted loci (67% for *TACRsgRNA1* and 91% for *TACRsgRNA2*, column three). In these reads, there was a high level of sequence variation in the targeted locus (S3 Table). Thus, we manually filtered through the unique reads generated by this analysis for each sgRNA (*TACRsgRNA1* and *TACRsgRNA2*) to distinguish whether or not they contained the prevalent IN/DEL. We then assessed the incidence of mutagenesis within the unique reads that did not contain the IN/DEL (17% for *TACRsgRNA1* and 18% for *TACRsgRNA2*, column six). These percentages align with the penetrance observed when both sgRNAs were used for CRISPR (Fig 7). We suspect that the high incidence of the IN/DEL may reflect a change in the *C. robusta* population used for this follow-up assay. This sequencing assay was conducted using gametes from adults harvested from a Los Angeles population that is distinct from population used in our CRISPR experiments. As expected for targeted mutagenesis, all detected deletions that potentially resulted in a frameshift occurred within 60 base pairs of the protospacer-adjacent motif (PAM) with a peak that occurred 3 base pairs upstream of the PAM (S3 Table).
(TIF)

**S1 Movie. *PC2>Cas9* control D10 juvenile exhibiting stereotypical peristalsis.** Juvenile expressing *PC2>Cas9* and an sgRNA targeting *GFP*.
(MOV)

**S2 Movie. *PC2>Cas9* TK crispant D10 juvenile exhibiting a pericardial bubble with no discernible myocardial tube.** Juvenile expressing *PC2>Cas9* and a pair of sgRNAs targeting *TK*.
(MOV)

**S3 Movie. *Mesp>Cas9* control D10 juvenile exhibiting stereotypical peristalsis.** Juvenile expressing *Mesp>Cas9* and an sgRNA targeting *GFP*.
(MOV)

**S4 Movie. *Mesp>Cas9* TACR crispant D10 juvenile showing dramatically disrupted myocardial tube along with abnormal peristalsis.** Juvenile expressing *Mesp>Cas9* and a pair of sgRNAs targeting *TACR*.
(MOV)

**S1 Table. Detailed list of analyzed genes.** Column 1 lists gene names. Column 2 lists 2012 KH IDs. Column 3 lists the cluster in which each analyzed gene was predominantly expressed. Column 4 lists putative cell types to which each cluster relates.
(CSV)

**S2 Table. Heart clusters matched to blood cell states.** Column 1 lists Leiden Clusters identified as heart cell types. Column 2 lists Leiden Clusters identified as blood cell types. Column 3 lists the blood cell types associated with the Leiden Cluster listed in Column 2.
(CSV)

**S3 Table. Unique reads for TACRsgRNA1 and TACRsrRNA2 IN/DEL analysis.** Genewiz sequencing was used to analyze partial penetrance of sgRNAs.
(XLSX)

**S1 Data. Raw data underlying graphs found in** Figs 1K, 4B, 4C, 4D, 4E, 4G, 5D, 5I, 5L, 5P, 7B, **S7A, and S7H.** A single file consisting of multiple tabs, each of which correspond to the data underlying a particular graph.
(XLSX)

## Acknowledgments

The neural transgenic lines were supported by the National BioResource Project, Japan. The authors would like to thank Alberto Stolfi and Eduardo Gigante for their input during the conceptualization of this study. H.N.G. thanks M.B. for their support.

## Author contributions

**Conceptualization:** Hannah N. Gruner, C. J. Pickett, Bradley Davidson.

**Data curation:** Hannah N. Gruner, C. J. Pickett.

**Formal analysis:** Hannah N. Gruner, C. J. Pickett, Jasmine Yimeng Bao, Tal D. Scully, Shaoyang Ning, Bradley Davidson.

**Funding acquisition:** Bradley Davidson.

**Investigation:** Hannah N. Gruner, C. J. Pickett, Jasmine Yimeng Bao, Richard Garcia, Akiko Hozumi, Mavis Gao, Gia Bautista, Keren Maze, HaEun Karissa Lim, Tomohiro Osugi, Maeve Collins-Doijode, Ofubofu Cairns, Gabriel Levis, Shu Yi Chen, TaiXi Gong, Honoo Satake, Yasunori Sasakura.

**Methodology:** Hannah N. Gruner, C. J. Pickett, Jasmine Yimeng Bao, Richard Garcia, Shaoyang Ning, Allon Moshe Klein, Yasunori Sasakura, Bradley Davidson.

**Project administration:** Bradley Davidson.

**Resources:** Yasunori Sasakura.

**Supervision:** Bradley Davidson.

**Validation:** Hannah N. Gruner, C. J. Pickett, Tal D. Scully, Sydney Popsuj, Allon Moshe Klein, Bradley Davidson.

**Visualization:** Hannah N. Gruner, C. J. Pickett, Akiko Hozumi, Mavis Gao, Gia Bautista, Keren Maze, HaEun Karissa Lim, Maeve Collins-Doijode, Yasunori Sasakura.

**Writing – original draft:** Hannah N. Gruner, C. J. Pickett, Bradley Davidson.

**Writing – review & editing:** Hannah N. Gruner, C. J. Pickett, Sydney Popsuj, Yasunori Sasakura, Bradley Davidson.

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
