## [Editor Report · Decision Letter 0]

5 May 2025

Dear Dr Davidson,

Thank you for submitting your manuscript entitled "Neural signaling contributes to heart formation and growth in the invertebrate chordate, Ciona robusta" for consideration as a Research Article by PLOS Biology.

Your manuscript has now been evaluated by the PLOS Biology editorial staff, as well as by an academic editor with relevant expertise, and I am writing to let you know that we would like to send your submission out for external peer review.

Once your full submission is complete, your paper will undergo a series of checks in preparation for peer review. After your manuscript has passed the checks it will be sent out for review. To provide the metadata for your submission, please Login to Editorial Manager (https://www.editorialmanager.com/pbiology) within two working days, i.e. by May 07 2025 11:59PM.

Kind regards,

Taylor

Taylor Hart, PhD,

Associate Editor

PLOS Biology

thart@plos.org

---

## [Decision Letter · Decision Letter 1]

5 Jul 2025

Dear Dr Davidson,

Thank you for your patience while your manuscript "Neural signaling contributes to heart formation and growth in the invertebrate chordate, Ciona robusta" was peer-reviewed at PLOS Biology. It has now been evaluated by the PLOS Biology editors, an Academic Editor with relevant expertise, and by several independent reviewers.

In light of the reviews, which you will find at the end of this email, we would like to invite you to revise the work to thoroughly address the reviewers' reports.

You'll see that the reviewers find the results compelling and think that the study employs robust methods. However, two of the reviewers also note some claims that are not fully supported by the experiments presented, areas requiring further clarification, and various inconsistencies. In light of these reports, we would like to invite you to submit a revision that addresses all the issues raised by the reviewers. In particular, the revised submission should include additional work to clarify the role of tachykinin signaling in heart development in line with Reviewer 2's comments. The revision should also include textual changes to improve the overall clarity and maintain a broad perspective on heart development appropriate for non-specialist readers.

Given the extent of revision needed, we cannot make a decision about publication until we have seen the revised manuscript and your response to the reviewers' comments. Your revised manuscript is likely to be sent for further evaluation by all or a subset of the reviewers.

**IMPORTANT - SUBMITTING YOUR REVISION**

*Re-submission Checklist*

*Published Peer Review*

*PLOS Data Policy*

*Blot and Gel Data Policy*

Sincerely,

Taylor

Taylor Hart, PhD,

Associate Editor

PLOS Biology

thart@plos.org

REVIEWS:

Reviewer #1: The manuscript titled "Neural signaling contributes to heart formation and growth in the invertebrate chordate, Ciona robusta" by H.N Gruner and colleagues represents a thorough investigation of the heart formation in post-metamorphosis Ciona, an invertebrate chordate belonging to the closest living relative to vertebrates. Its study will enhance our understanding of the vertebrate heart's evolution and potentially uncover new therapeutic avenues for heart regeneration. The authors used a combination of lineage tracing by both photoconversion and EdU pulse-chase labeling, single-RNA seq, reporter assays, in situ hybridization, chemical inhibition, and CRISPR/Cas9 mutagenesis to decipher the role of a structure named "undifferentiated line (UL)," Wnt signaling and innervation in Ciona heart formation. They found that cardiomyocyte progenitors are localized in the UL in juveniles. These progenitors first divide asymmetrically in its distal part, one cell staying in this cluster while the other moves toward the midline of the UL, where they can divide either symmetrically or asymmetrically, leaving the midline to differentiate into cardiomyocytes. The authors then performed single-cell RNA sequencing of the adult heart and identified clusters corresponding to cell types composing the heart, including its innervation. They followed up these assays by analyzing the role of Wnt signaling in heart formation, finding that its inhibition increases UL proliferation while its overactivation inhibits UL proliferation. They then characterized the intrinsic and extrinsic innervation of the heart as well as the pacemaker cells of the heart. They identified both cholinergic and dopaminergic intrinsic neurons. However, they do not seem to affect heart rate. They then studied the role of tachykinin since some of these neurons express tachykinin receptors. They discovered that tachykinin promotes the proliferation of UL distal clusters using receptor antagonists and mutagenesis of the tachykinin gene in the cerebral ganglion and the tachykinin receptor gene in the heart lineage.

The authors used robust experimental methods. Their findings are compelling and well-presented. The manuscript is also easy to read and includes numerous cartoons to help non-familiar readers understand the anatomy and development of Ciona's heart, a non-conventional model.

However, the current manuscript has issues that need to be addressed:

* The authors proposed that the juvenile heart has cardiac precursors dividing symmetrically after leaving the UL and before differentiating into cardiomyocytes. While it is an appealing hypothesis, the authors do not have direct evidence. They observed cell pairs in Figure 2H. These cells might divide symmetrically in the UL before leaving instead of after. Moreover, they do not seem Edu-positive. Did the authors ever observe EdU-positive cells in the myocardium of juveniles after the 30 minute EdU-pulse (lines 326-329)? The author might expect the event to be as frequent as the asymmetric midline UL division (lines 351-352). Why are the authors not considering the distal clusters as the most undifferentiated/stem-like cells and the midline cells as the "transit-amplifying zone" in juveniles? Furthermore, cell division perpendicular to the UL midline with one daughter cell not anymore in contact with the anterior side of the UL is not a sign of asymmetric cell division since both MUV1B and MUV1A (Figure 2I and 2J) stay in the UL in contradiction with line 957-958 of the discussion. While the single-cell data is from the adult heart, building the transcriptional trajectory from the UL to the cardiomyocytes would be interesting to see if a cell state corresponding to cardiac precursor cells is present. In this regard, mapping the different UL subclusters might also be informative. In addition, could the cells with a distinctive morphology in the myocardium be a different cell type, such as immune cells?

* Plots displaying the data do not always match their statistical test:

o In panels 4B to 4D, stacked bar charts represent qualitative/categorical data. The authors should use the Chi-squared test of independence to test the statistical difference. T-test should be used to test the statistical difference between two samples of a normally distributed quantitative variable. Quantitative data is represented by dot plots, box plots, or violin plots, showing the distribution of the samples, as shown in Figures 5E and 5J. The authors should use a Mann-Whitney U/Wilcoxon rank sum test if the data are not normally distributed. For multiple comparisons (more than two groups), the authors should first perform a One-Way ANOVA followed by a t-test with a correction for multiple testing (such as Bonferroni). Moreover, if the authors have the cell quantification, it is better to display cell numbers instead of proportions.

o Panel 4G plot also displays a categorical variable (presence or not of more than the average + one standard deviation). A t-test should not be used.

o In panel 7b, the violin plot should be for continuous quantitative data. (Stacked) bar charts show proportion. Fisher's exact test is then correct.

o Was correction for multiple testing applied for the tests in panel 5J, S2A? Are the data normally distributed?

* The Neural/Pacemaker (NP) subclusters are characterized in detail. However, it deserves some clarification.

o How many cells were assigned to the NP cluster and then to the different subclusters?

o The heatmap in panel 3I shows that NP-B is both Chat- and Th-positive. However, the authors do not observe double-positive cells with reporter assays for Th and Vacht. What is the expression level of Th, Chat on the UMAP projection of the NP cluster?

o Panel S9B shows Vacht- and Pc2-double-positive cells. Is the co-expression a transient developmental effect? How is Pc2 expression on the UMAP projection of the NP cluster?

o Are NP-A expressing any marker for a given neuron type, such as GABAergic or glutamatergic?

o The authors also argue that NP-B represents a mix of neural and pacemaker cells derived from the mesodermal lineage. However, NP cells in Figure 3D do not express Tnni1 (Troponin) or any of the cardiomyocyte markers. What is the Tnni1 expression level on the UMAP projection of the NP cluster? Are the panels S8D showing double reporter assay for Vacht and Troponin maximum intensity projections? Where are these pacemaker cells in the cartoon in panel S8A? Strikingly, in vertebrates, pacemaker cells are not neurons but specialized cardiomyocytes concentrated in specific zones. They are hyperpolarized and able to generate an action potential. These cells are innervated by cholinergic neurons, among other neuron types. Are the authors hypothesizing that pre-vertebrate cholinergic pacemaker cells (one NP-B subtype) subfunctionalizes to give rise to vertebrate pacemaker cells? Neurons in mice also expressed Hcn4. Why would the cells be cholinergic? Where are cholinergic receptors expressed in the heart UMAP? Moreover, the instances of Phox2b-positive mesodermal-derived cells are not convincing. Phox2b is a cytoplasmic reporter (Unc26). Nevertheless, it seems nuclear when it is co-expressed with the Mesp reporter. Might this signal be autofluorescence associated with dead cells?

o Since the adult heart is still growing, could the NP-C subcluster represent a precursor state of NP-B?

* Based on the fluorescence in situ hybridization (Figure S12), it is possible that Tacr is more broadly expressed at the juvenile stage than in the adult. Neuropeptides usually rely on volume transmission and not on precise synaptic transmission. They might diffuse more than 100µm from their source. It might explain the dramatic phenotype observed upon Tacr CRISPR using Cas9 driven by Mesp regulatory sequences, matching the phenotype observed in Tk CRISPR juvenile. Tk would then directly provide pro-proliferative signals to UL.

Additional comments:

* The number of analyzed samples is not always mentioned.

* Line 568, 588, 620, 670, 684, 786, 788, 856 1012 reference the wrong figure.

* Scale bars are missing in panels 2A, 2J, Figure 5, panels 7A, 7E, 7F, S1A, S1B, S1C, S1D, S1E, S8B, S8C, S8E, S8F, S8D'' (named S8D'), Figure S10, panel S11B, S11G, S11H, Figure S13

* In the S1A legend, the authors mentioned XY projection. It is an XZ projection in the figure.

* Gene names are not always consistent; for example, Gata4/5/6 (line 457) instead of Gata.a and CrUL1 in the main text is named UL1 in Table S1 and Panel 7I and 7J or UL-1 in Figure 3.

* Gene and plasmid formats or names are not uniform. For example, the format of gene names in Figure 3 or Mesp is in uppercase, while Phox2b is capitalized (line 1856 and Figure S10). Some fusion proteins have one ":" others two (line 1856 and Figure S10).

* Wnt signaling might promote differentiation. In this case, its inhibition blocks differentiation. The cells stay as progenitors and keep proliferating. The effect of Wnt on proliferation at this stage might be indirect.

* Neural-like cells could be referenced as neural cells.

* Annotate the heart outline in Figures S8B and S8C.

* Phox2b is enriched in both NP-B and NP-C, not NP-A (Line 699)

* The drug Spantide II is referenced as Spantide I in the legend of Figure 5 and Spantide in the figure.

* Some references miss the journal name, or their abbreviation is not consistent.

In summary, this manuscript represents critical steps into our comprehension of heart development in Ciona juveniles, which will enhance our understanding of the origin of vertebrate cardiac innovation, including its intrinsic innervation. I support its publication in PloS Biology, contingent upon the recommended modifications.

Reviewer #2: In this manuscript the authors use the chordate model Ciona robusta to investigate post larval heart development. Specifically they are interested in neuronal signals that regulate heart growth. The effects on cell proliferation and heart growth are examined after manipulation of the canonical Wnt pathway using pharmacological tools, or CRISPR to inhibit tachykinin neuropeptide and tachykinin receptors. They propose that insights gained may be useful for understanding vertebrate heart growth.

I appreciate the amount of work that has gone into this manuscript. However, I found the paper unwieldy, the number of supplementary figures (13) and figures (7) is unbalanced and the different parts could be better integrated. The introduction is focused on mammals and some other vertebrates (which are often not identified for the reader). Overall, this introduction does not link well with the subject being studied, and it is also not picked up in the discussion. Much of the discussion is focussed on what they aim to do next rather than trying to make sense of their observations.

The first two figures describe the microscopic anatomy of the adult C. robusta heart and of the juvenile heart. While this would be of interest for a specialist reader, it is difficult to follow the evidence supporting the cartoon overview models shown. These cartoon models are Fig 1I for a presumptive transit amplifying zone; and Fig 2I for the undifferentiated line (UL) and myocardial precursor cell divisions. The challenge is the very low proliferative index and UL cell divisions were rarely observed. The core premise is therefore not convincing, line 387: "our data strongly supports the following core premise:..."

line 396, there is no citation for the protocol from the Klein lab; citation number [87] is incomplete

The transcriptomic analysis provides several clusters of cells for two stages of adult hearts. They focus on a cluster (no. 7) representing cardiomyocyte progenitors of the UL and validate expression of one selected gene, CrUL1, by in situ, S7 Fig. The pattern observed was however different to that of their reference gene, Gata.a. This is confusing and was not commented on any further.

The data in Figure 4 shows clearly that pharmacological inhibition of the canonical Wnt pathway leads to an increase in the frequency of cardiomyocyte progenitor division. This is interesting and unexpected. They next show that pharmacological inhibition of tachykinin peptide signalling reduces the number of progenitors. This is somewhat supported by knockdown of tachykinin or its receptor using CRISPR-Cas9 approaches, which are mosaic and lead to a variable phenotype with partial penetrance. In some cases the formation of the myocardial tube is affected.

The relationship between canonical Wnt and tachykinin pathways is not clear and not explored further and discussion is very limited to: line 1017 "We are also interested in investigating whether TK signaling instructs neural-like cells to secrete cWnt inhibitors."

Given that the introduction is focussed on mammals, the authors should consider how their findings are relevant to vertebrates. Alternatively, change the focus of the introduction.

Other issues:

line 684, "these distinct neurons may be represented by sub-clusters NP-A or NP-B respectively (Fig 2I)." It is not clear why the authors refer to this cartoon model here.

line 720-722, "Application of the TH inhibitor α-methyl-para-tyrosine (AMPT) lowered heart rate slightly, but this result was not significant. In contrast, application of the acetylcholine agonist acetylcholine chloride and the TACR inhibitor aprepitant had no discernible impact on heart rate." This does not make sense. Why do the authors say "in contrast" when in both cases there was no effect.

line 939, "we posit that both distal and midline progenitors behave like cardiac stem cells, dividing asymmetrically to self-renew while producing a midline or precursor daughter respectively (Fig 2I). We are currently working to identify genetic markers that distinguish between these two progenitor populations." Again as above, I am not convinced that there is sufficient evidence to support this.

Reviewer #3: Review of the manuscript by Gruner et al.

The paper by Gruner et al. provides novel insights into how neural signals are involved in the development of heart cells during Ciona robusta development. The manuscript is well written, and the experiments are comprehensive and clearly described. The results are thoroughly discussed and are of significant interest to the community.

The authors begin with a detailed analysis of the cellular composition of the heart in both juvenile and adult Ciona, with a particular focus on the undifferentiated lineage (UL). The proliferative potential of different cell populations was assessed using EdU pulse-chase experiments, revealing a specific pattern of mitotic activity in both juveniles and adults. Specific asymmetric division in juvenile UL support the conclusion that the UL represents a source of undifferentiated, proliferative cells.

Single-cell transcriptomic analysis of isolated heart tissue identified distinct subpopulations of heart cells. Notably, the authors identified a population of cells putatively associated with neural-like cell types through scRNA-seq analysis. Using stable transgenic lines and a photoconvertible marker (KAEDE), they were able to trace, characterize, and investigate the potential role of these cells during heart formation.

The authors make appropriate use of a wide range of experimental approaches, allowing them to address the scientific question from multiple perspectives. Functional studies using CRISPR/Cas9 gene editing and pharmacological inhibitors revealed key roles for Wnt signaling and tachykinin in regulating the growth and differentiation of heart cells.

Overall, this study represents an important contribution to the field highlighting Ciona robusta as a powerful model for studying the interplay between neural signals and cardiac development.

---

## [Decision Letter · Decision Letter 2]

22 Jan 2026

Dear Dr Davidson,

Thank you for your patience while we considered your revised manuscript "Neural signaling contributes to heart formation and growth in the invertebrate chordate, Ciona robusta" for publication as a Research Article at PLOS Biology. This revised version of your manuscript has been evaluated by the PLOS Biology editors, the Academic Editor, and two of the original reviewers. We apologize for the delay as we continue to work through the backlog from the recent holidays.

Based on the reviews, we are likely to accept this manuscript for publication, provided you satisfactorily address the remaining points raised by the reviewers and the Academic Editor:

After further discussion with the Academic Editor, we think that the Introduction and Discussion sections of the paper can be tightened further (an aspect that Reviewer 2 indicated was 'somewhat' addressed in the most recent revision). We request that you streamline these aspects of the writing to focus on the points that are most relevant for your specific study rather than providing a general overview of the field. We also think that adding smoother transitions between paragraphs would help.

Please also make sure to address the following data and other policy-related requests.

IMPORTANT: Please ensure that you address all of the following editorial requirements:

**Data:

We require several items related to this -- please attend to each of the points below.

-- RNAseq data should be uploaded to a standard repository and made available.

-- Please supply the numerical values either in a supplementary excel file or as a permanent DOI’d deposition for the following figures:

Figs:

1K

4E

5DIJLP

7B

S7AH

-- Please cite the location of the data clearly in all relevant main and supplementary Figure legends, e.g. “The data underlying this Figure can be found in S1 Data” or “The data underlying this Figure can be found in https://doi.org/10.5281/zenodo.XXXXX”

-- Supplementary files (e.g., excel). Please ensure that all data files are uploaded as 'Supporting Information' and are invariably referred to (in the manuscript, figure legends, and the Description field when uploading your files) using the following format verbatim: S1 Data, S2 Data, etc. Multiple panels of a single or even several figures can be included as multiple sheets in one excel file that is saved using exactly the following convention: S1_Data.xlsx (using an underscore).

-- Please ensure that your Data Statement in the submission system accurately describes where your data can be found and is in final format, as it will be published as written there.

**Supplementary movies:

-- We see that you included Google drive links to the movies, but we could not access them. Please upload the movies as supplementary information files, or provide them in a stable online repository, preferably with a DOI.

We expect to receive your revised manuscript within two weeks.

*Published Peer Review History*

*Press*

Sincerely,

Taylor

Taylor Hart, PhD,

Associate Editor

thart@plos.org

PLOS Biology

Reviewer remarks:

Reviewer #1: The authors did an excellent job on their revised manuscript, reinforcing their findings with orthogonal approaches. They streamlined their narrative and focused on their most important findings. They also addressed most of my comments and concerns. A few discrepancies persist, which is not unusual given the scope of the work:

- In the caption of Figure 4 (lines 588-589), p-values are mentioned as asterisks. In the figure, they are in numbers.

- What statistical test was done for Figure 4G? I could not find it in the caption or the method. I still think that it would be easier to understand a (violin/dot) plot with the UL cell number for this experiment if the authors have the exact cell number.

- In lines 726 and 727, the author mentioned to panel 5E and not 5D.

- In the caption of Figure 5P (line 760) and the statistic section of the methods, quantification of EdU+ cells is divided into dorsal, ventral, and midline UL, while the plot now shows distal and midline UL.

- The statistical tests in Figure 5I, 5L, 5P are Mann-Whitney in the caption, but t test in the statistics section of the methods.

- In the statistics section of the methods (line 1071), the adult heart is divided into distal, middle, apical. In Figure 1K, it is divided into distal and middle.

- The caption of Figure 7B (line 881) mentions a violin plot, while it is a bar plot in the figure.

In summary, with its additional results, orthogonal approaches, and refined narrative, this study on heart formation and growth in Ciona juveniles by H.N. Gruner and colleagues strongly deserves its publication in Plos Biology.

Reviewer #2: The authors have thoroughly revised the manuscript and explained changes in a detailed reponse. The have tidied up the text somewhat and increased focus on the more pertinent aspects of the work. In the experiments aimed at clarifying the role of tachykinin signaling in heart development, they have commented on the partial penetrance of the phenotype. Other queries have also been dealt with satisfactorily.

Some of the panels in figure 7 contain (?) (7D, F, H, J). I could not see where this is explained in the text or the legend. Sorry - I missed this before.

The references are not always correctly formatted, e.g. refs 1,3,5, 12,14, 15, 23 have no Journal, ref 48 has no title

---

## [Editor Report · Decision Letter 3]

27 Feb 2026

Dear Dr Davidson,

Thanks for submitting the latest revision of your manuscript. However, we require a few additional changes before we can move forward. Can you please look into the following items?

- R2R: As your paper went through a second round of peer review, we expected to find a response to the latest comments from Reviewers 1 and 2. We understand that the points raised were minor, but we would still appreciate if you could provide a response indicating how you addressed these points.

- Data availability: We tried to access the link to your data uploaded on NCBI, but we got following message: "Error: Unauthorized" . Please make sure that this is accessible, and also ensure that the statement is finalized for publication.

- Data Availability Statement: The current version of your statement in the form does not mention the Supporting Information files. Please revise the statement to mention where all data related to your paper can be found, and make sure that it conforms to any changes you make to address the following points.

- Supporting Information: Two of the figure legends mention S1 Data, but there is no uploaded file with this name.

- Supporting Information: Some of the figure legends include a link to a Google sheet containing numerical data. Perhaps this was meant to be S1 Data? Please either upload this to a stable repository with a DOI, or preferably upload these data as a Supporting Information excel file (you can include multiple tabs pertaining to different figure panels, as you have done in the Google sheet). If you choose this option, you can call it S1 Data (S1_Data.xlsx), and include the title and caption for it in the manuscript document after the supplementary tables, and make sure that it is referred to consistently in all relevant figure legends.

We expect to receive your revised manuscript within a week.

*Published Peer Review History*

*Press*

Sincerely,

Taylor

Taylor Hart, PhD,

Associate Editor

thart@plos.org

PLOS Biology

Reviewer remarks:

Reviewer 1

The authors did an excellent job on their revised manuscript, reinforcing their findings with orthogonal approaches. They streamlined their narrative and focused on their most important findings. They also addressed most of my comments and concerns. A few discrepancies persist, which is not unusual given the scope of the work:

- In the caption of Figure 4 (lines 588-589), p-values are mentioned as asterisks. In the figure, they are in numbers.

- What statistical test was done for Figure 4G? I could not find it in the caption or the method. I still think that it would be easier to understand a (violin/dot) plot with the UL cell number for this experiment if the authors have the exact cell number.

- In lines 726 and 727, the author mentioned to panel 5E and not 5D.

- In the caption of Figure 5P (line 760) and the statistic section of the methods, quantification of EdU+ cells is divided into dorsal, ventral, and midline UL, while the plot now shows distal and midline UL.

- The statistical tests in Figure 5I, 5L, 5P are Mann-Whitney in the caption, but t test in the statistics section of the methods.

- In the statistics section of the methods (line 1071), the adult heart is divided into distal, middle, apical. In Figure 1K, it is divided into distal and middle.

- The caption of Figure 7B (line 881) mentions a violin plot, while it is a bar plot in the figure.

In summary, with its additional results, orthogonal approaches, and refined narrative, this study on heart formation and growth in Ciona juveniles by H.N. Gruner and colleagues strongly deserves its publication in Plos Biology.

Reviewer 2

The authors have thoroughly revised the manuscript and explained changes in a detailed reponse. The have tidied up the text somewhat and increased focus on the more pertinent aspects of the work. In the experiments aimed at clarifying the role of tachykinin signaling in heart development, they have commented on the partial penetrance of the phenotype. Other queries have also been dealt with satisfactorily.

Some of the panels in figure 7 contain (?) (7D, F, H, J). I could not see where this is explained in the text or the legend. Sorry - I missed this before.

The references are not always correctly formatted, e.g. refs 1,3,5, 12,14, 15, 23 have no Journal, ref 48 has no title

---

## [Editor Report · Decision Letter 4]

3 Mar 2026

Dear Dr Davidson,

Thank you for the submission of your revised Research Article "Neural signaling contributes to heart formation and growth in the invertebrate chordate, Ciona robusta" for publication in PLOS Biology. On behalf of my colleagues and the Academic Editor, Anna Kicheva, I am pleased to say that we can in principle accept your manuscript for publication, provided you address any remaining formatting and reporting issues. These will be detailed in an email you should receive within 2-3 business days from our colleagues in the journal operations team; no action is required from you until then. Please note that we will not be able to formally accept your manuscript and schedule it for publication until you have completed any requested changes.

PRESS

Sincerely,

Taylor

Taylor Hart, PhD,

Associate Editor

PLOS Biology

thart@plos.org